# The coenzyme thiamine diphosphate displays a daily rhythm in the *Arabidopsis* nucleus

Zeenat B. Noordally[1], Celso Trichtinger[1], Ivan Dalvit[1], Manuel Hofmann[1], Céline Roux[1], Nicola Zamboni[2], Lucille Pourcel[1], Elisabet Gas-Pascual [1], Alexandra Gisler[1] & Teresa B. Fitzpatrick [1✉]

In plants, metabolic homeostasis—the driving force of growth and development—is achieved through the dynamic behavior of a network of enzymes, many of which depend on coenzymes for activity. The circadian clock is established to influence coordination of supply and demand of metabolites. Metabolic oscillations independent of the circadian clock, particularly at the subcellular level is unexplored. Here, we reveal a metabolic rhythm of the essential coenzyme thiamine diphosphate (TDP) in the *Arabidopsis* nucleus. We show there is temporal separation of the clock control of cellular biosynthesis and transport of TDP at the transcriptional level. Taking advantage of the sole reported riboswitch metabolite sensor in plants, we show that TDP oscillates in the nucleus. This oscillation is a function of a light-dark cycle and is independent of circadian clock control. The findings are important to understand plant fitness in terms of metabolite rhythms.

[1] Department of Botany and Plant Biology, University of Geneva, 1211 Geneva, Switzerland. [2] Institute of Molecular Systems Biology, ETH Zurich, 8093 Zurich, Switzerland. ✉email: theresa.fitzpatrick@unige.ch

Metabolic homeostasis is achieved through the spatio-temporal control of the associated network of enzymes, many of which are vitally dependent on coenzymes for activity. The circadian clock is established to influence the dynamic behavior of this system, in particular coordinating the supply of metabolites with demand. The coenzyme TDP, a form of vitamin $B_1$, is essential for cell energy supply in all organisms, as it plays a pivotal role in carbon metabolism. It is required for functionality of key regulatory enzymes of glycolysis, the citric acid cycle, the oxidative pentose phosphate pathway[1], carbon fixation through the Calvin cycle[2], and the non-mevalonate isoprenoid biosynthesis pathway, from which thousands of metabolites are derived including chlorophyll and several phytohormones in plants[3]. However, correlation between the levels of TDP and operational output of these pathways is unknown.

Elucidation of the biosynthesis de novo pathway of TDP in plants has been completed over the last several years. It comprises separate branches for the biosynthesis of the thiazole and pyrimidine moieties, by the action of THI1 and THIC, respectively, in the plastid (Fig. 1)[4–7]. Condensation of the two moieties is carried out by TH1 in the plastid to produce thiamine monophosphate (TMP) (Fig. 1)[8]. TMP is then dephosphorylated to thiamine and the final conversion to TDP, catalyzed by TPK, takes place in the cytosol, at least in *Arabidopsis* (Fig. 1)[9]. It is not known if a plastid-specific phosphatase exists for TMP, or if it is exported from the plastid and dephosphorylated in the cytosol. Although a TMP-specific phosphatase (TH2/PALEGREEN1) has been recently identified, it is predominantly localized to the mitochondria[10,11].

It is intriguing that TDP is made outside of the organellar powerhouses (plastids and mitochondria) that rely on it. One pertinent question therefore is how TDP, a polar molecule, is delivered to the organellar apoenzymes. However, identification of these transporters largely remains elusive and surprisingly has received little attention. To date, no transporter of TDP across plastid membranes has been identified. Although, very recently, the single member of the nucleobase cation symporter family in *Arabidopsis*, NCS1[12], also known as PLUTO[13], was reported to transport the pyrimidine moiety precursor, hydroxymethylpyrimidine[14]. NCS1 was previously shown to import pyrimidine nucleobases into plastids[12]. Conversely, transporters of TDP into mitochondria were identified almost two decades ago in yeast[15], and later in Drosophila[16], and humans[17] and are annotated *T*hiamine *P*yrophosphate *C*arriers (TPC). Homologs have been recognized in *Arabidopsis* and maize[18] named TPC1 and TPC2 (Fig. 1) and were shown to restore prototrophy to the *TPC1* null mutant of yeast and boost activity of acetolactate synthase[18]. However, these transporters have not been characterized in planta.

A second question relates to homeostasis of TDP itself and how biosynthesis and transport are integrated. The essential role of

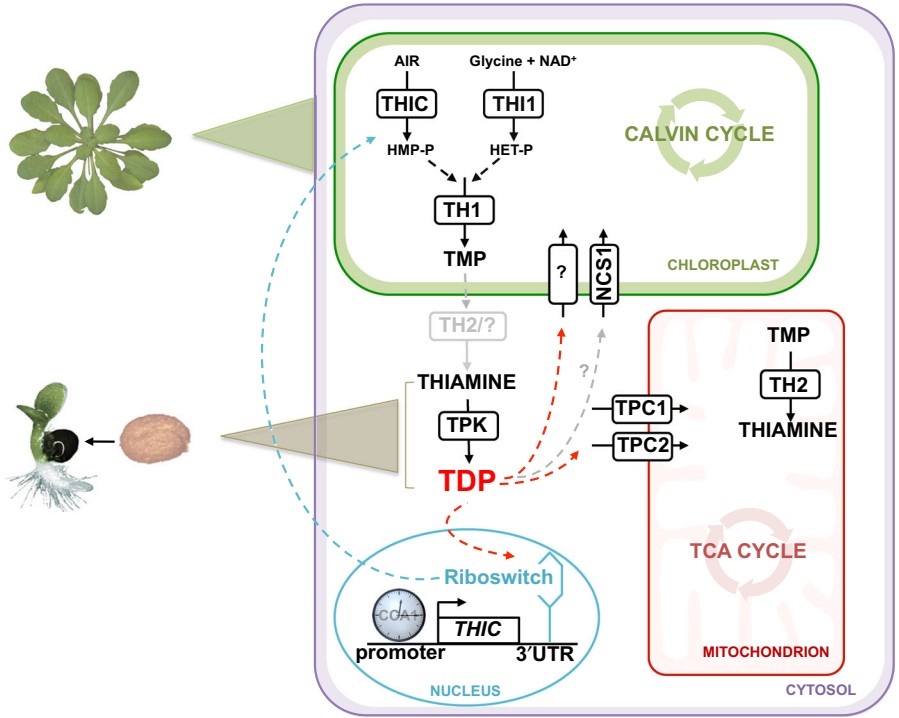

**Fig. 1 Overview of thiamine metabolism in plants.** Thiamine monophosphate (TMP) is biosynthesized de novo in the chloroplast by condensation of hydroxymethylpyrimidine-phosphate (HMP-P) and hydroxyethylthiazole-phosphate (HET-P), which are derived from 5-aminoimidazole ribotide (AIR), nicotinamide adenine dinucleotide ($NAD^+$) and glycine as precursors, followed by dephosphorylation and pyrophosphorylation to thiamine diphosphate (TDP) in the cytosol. The pathway involves the enzyme action of THIC, THI1, TH1, TPK and TH2, although the latter enzyme appears to be predominantly in the mitochondrion. TDP is needed for central metabolic pathways in organelles: the TCA cycle and the Calvin cycle, as well as glycolysis in the cytoplasm (not shown). While TPCs (TPC1 and TPC2) import TDP at the mitochondrial membrane, it is not known how TDP is transported into the chloroplast or nonphotosynthetic plastids, although the promiscuous transporter NCS1 is thought to import HMP or TDP (represented by the gray arrow and by "?"). In the nucleus, expression of the *THIC* gene is regulated by the circadian clock through a CCA1 binding site in its promoter and by the level of free TDP through a riboswitch present in the *3'-UTR*. The biosynthesis de novo pathway of TDP is active in green tissue about 5 days after germination (indicated by a green triangle), whereas TDP is supplied through the action of TPK on thiamine in germinating seeds (indicated by a brown triangle). The blue arrow represents negative feedback regulation of HMP-P production caused by the binding of TDP to the *THIC* riboswitch in the nucleus. The red arrows represent the necessity to transport TDP into the organelles.

TDP in the energy-generating reactions of the cell means that supply needs to be tightly regulated, as perturbance may have a strong impact. For example, limited TDP leads to only a minor decrease in transketolase activity, yet a strong perturbation of carbon metabolism is incurred impairing plant growth[19]. Comparably, overexpression of transketolase results in TDP auxotrophy-like symptoms, as other enzymes are divested of the coenzyme, and negatively impacts plant health[20]. We must also consider that metabolite levels need to be temporally controlled to be in tune with the diurnal needs of the plant. For example, carbon metabolism needs to be coordinated such that sugars are stored during the day to be used for metabolic activities at night, and is reported to be tightly integrated with the circadian clock[21]. As a corollary, it follows that TDP supply needs to be tightly coordinated with the needs of enzymes dependent on it as a coenzyme. Thus far, there have been few reports on the regulation of TDP levels in plants with the predominant exception of the mechanism of control of *THIC* expression via the RNA aptamer in its 3′-UTR, termed a riboswitch (Fig. 1). Binding of TDP to the *THIC* riboswitch induces alternative splicing that leads to an unstable mRNA, mediating negative feedback regulation on TDP biosynthesis de novo[22,23]. Studies in *Arabidopsis* have also shown that the expression of the *THIC* gene is regulated by the circadian clock likely through a CCA1 binding site in its promoter (Fig. 1)[24]. Further studies on regulation of TDP levels in plants could provide an important paradigm for coenzyme supply and demand in daily organismal fitness.

Here, in addition to demonstrating the importance of TDP for central metabolism and thereby growth and development, we report on components of the intracellular transport of TDP and/ or its precursors in the plant model *Arabidopsis*. We show that both TDP biosynthesis and transport are transcriptionally regulated by the circadian clock. Importantly, biosynthesis is temporally separated from transport and is likely an important parameter for coordination of TDP supply and demand. We have used the alternative splicing of the *THIC* riboswitch as a natural sensing tool to report on daily modulation of TDP levels. Indeed, levels of TDP oscillate at the intracellular level (i.e. the nucleus) peaking in abundance in the first half of the day. This peak in abundance is derivative of light-dark cycles that is independent of the circadian clock regulation.

## Results

**TDP is essential for central metabolism.** During seed development/maturation in plants, the nonphosphorylated $B_1$ vitamer thiamine is stored, and upon germination is converted to TDP (Fig. 1)[25], furnishing enzymes of central metabolism in the cotyledons. Beyond this however, the development of true leaves requires biosynthesis de novo of TDP[7]. This is corroborated by the fact that thiamine biosynthesis de novo mutants, such as *thiC*, cannot proceed beyond the cotyledon stage of development[7]. The necessity of having biosynthesis de novo can be bypassed by supplementation with thiamine[7]. Bypassing biosynthesis de novo in these previous experiments was performed by supplementing with excessive levels of thiamine (up to 100 μM). In order to test whether core metabolism is indeed dependent on TDP levels, we grew the *thiC* mutant in the presence of a range of thiamine concentrations (0−1.5 μM), spanning physiologically measured levels. Supplementation with thiamine had little effect on wild-type growth (Fig. 2a). However, the growth of *thiC* beyond the cotyledon stage was dependent on the level of thiamine supplementation (Fig. 2a). Moreover, TDP levels in these seedlings showed a clear correlation with thiamine supplementation levels (Fig. 2b). Furthermore, we utilized a nontargeted metabolomics approach to assess metabolite changes as a function of the

thiamine supplementation level. Mapping the set of detectable metabolites onto a scheme of central metabolism illustrates the panorama of metabolite changes with and without thiamine supplementation in *thiC* (Supplementary Fig. 1a). Such changes are not seen in the corresponding wild type under the same conditions (Supplementary Fig. 1b). Across the range of metabolites measured, a core set of metabolites was seen to be significantly dependent on the level of thiamine supplementation (Fig. 2c). Notably, many of these metabolites were photosynthetic pigments. From the measurements of TDP levels in these seedlings, it appears that ca. 1 μmol g$^{-1}$ FW (at the tissue level) is required for growth to approach wild-type levels (Fig. 2a, b), because stunting and chlorosis can still be observed with lower supplementation levels (Fig. 2a). These data therefore highlight the fact that TDP is needed in relatively small amounts but it is also evident that even a small deficit (low nM) severely reduces growth, likely by directly impacting photosynthesis, essentially halting development. The data thus demonstrate that TDP supply is essential for central metabolism and limitation of this coenzyme has major impacts on growth and development.

**The mitochondrial transporters *TPC1* and *TPC2* are essential.** As TDP is an essential metabolite and its biosynthesis is well mapped out by now, we next turned our attention to the dearth of knowledge in relation to the characterization of its transport particularly at the subcellular level. Although *Arabidopsis* TPC1 (At3g21390) and TPC2 (At5g48970) have been shown to be involved in TDP transport through yeast-based assays[18], their relevance was not investigated in plants. Note, *Arabidopsis* TPC1 and TPC2 should not be confused with the *Arabidopsis* two-pore channel (TPC) protein at locus At4g03560. To investigate the functional significance of *TPC1* and *TPC2* in planta, seeds of putative mutant lines were obtained from the available collections (Supplementary Fig. 2a). In particular, independent T-DNA insertion lines SAIL_130_D09 (annotated *tpc1-1*) and GK-236B06 (annotated *tpc1-2*), as well as GK-870B10 (annotated *tpc2-1*) and SAIL_127_G03 (annotated *tpc2-2*) were analyzed. In all cases, the T-DNA insertion lines could be isolated to homozygosity. PCR analysis established insertion of the T-DNA in intron 3 of SAIL_130_D09 (bp 1759), exon 6 of GK-236B06 (bp 2623), intron 2 of GK-870B10 (bp 971) and intron 3 of SAIL_127_G03 (bp 1302) (Supplementary Fig. 2a). To assess for an incongruent phenotype, plants were grown in soil under standard conditions. However, no morphological defects could be deciphered compared to the wild type (Col-0) grown under the same conditions (Supplementary Fig. 2b). Indeed, although transcript levels of *TPC1* and *TPC2* is severely reduced in respective mutant lines (Supplementary Fig. 2c), *TPC1* transcript levels in *tpc2* and *TPC2* transcript levels in *tpc1* did not differ significantly from their wild-type (Col-0) levels in the respective mutant partner line (Supplementary Fig. 2c). We also examined the $B_1$ vitamer levels (thiamine, TMP and TDP) in mutant lines by high performance liquid chromatography, which were similar to those in the corresponding wild type grown under the same conditions (Supplementary Fig. 2d). A tissue expression analysis indicates that both genes are ubiquitously expressed (Supplementary Fig. 2e), and parallels publicly available transcriptomic datasets in Genevestigator[26]. We next performed reciprocal crosses of *tpc1-1* and *tpc2-1*. In both cases, while seedlings hemizygous for each mutation could be obtained, no mutant homozygous for the double *tpc1 tpc2* mutation could be isolated. Interestingly, a segregation analysis of the F1 progenies from the crosses based on the BASTA™ and sulfadiazine resistance of *tpc1-1* and *tpc2-1*, respectively, revealed a distortion of the classic Mendelian inheritance expected (i.e. 1:1 resistance:sensitivity) for

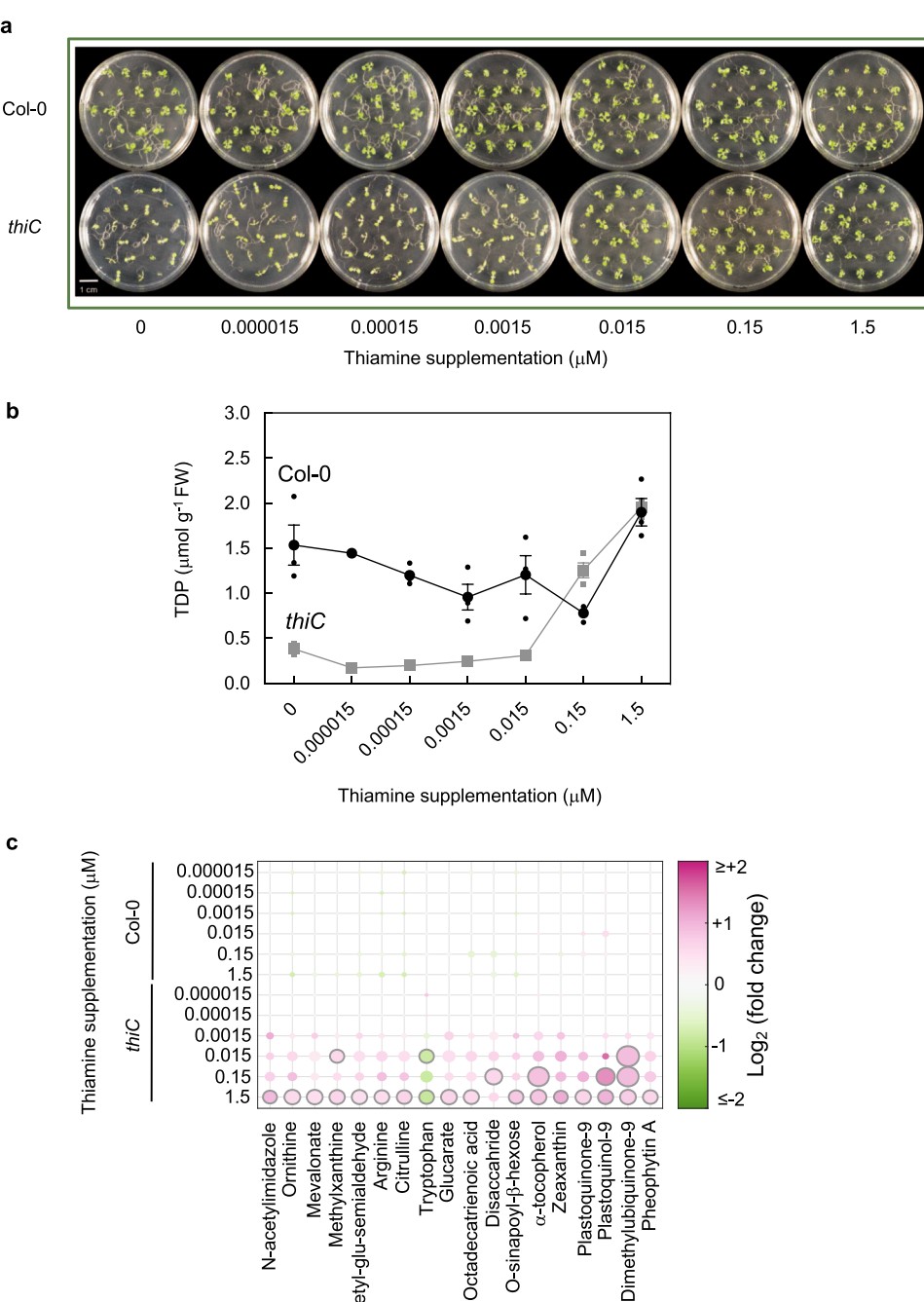

**Fig. 2 Central metabolism depends on thiamine supply. a** Photographs of 11-day-old wild type and *thiC Arabidopsis* seedlings supplemented with thiamine levels as indicated. Seedlings were grown on 1/2 MS agar plates under a 16-h photoperiod (120 μmol photons m$^{-2}$ s$^{-1}$) at 22 °C and 8 h of darkness at 18 °C. **b** Levels of thiamine diphosphate (TDP) measured in seedlings of wild type and the *thiC* mutant as shown in (**a**) as a function of the thiamine supplementation level as indicated. Data of three individual biological replicates of pooled material ($n = 15$) is shown with error bars representing SE. **c** Significant metabolite content changes as a function of thiamine supplementation (as in (**a**)) using nontargeted metabolomics. Sphere size and color shade correlate with the extent of change of the particular metabolite as indicated. The plot reports all metabolites out of 372 detected that were found to pass significance criteria (|log2(fold-change)| > 0.5 and adj. *p* value < 0.01) in at least one condition. The data shown are relative to the control with no thiamine supplementation. Significant points are encircled in gray.

a sole embryo-based problem (Supplementary Table 1). This was confirmed by a chi-square test, which rejected the hypothesis of a sole embryo defect at a significance level of $P < 0.05$ and suggests that a germline defect may also be involved. Effects such as a low penetrance gametophytic problem or silencing of the resistance selection may also account for these observations. Nonetheless, we conclude that the combination of *TPC1* and *TPC2* is essential for *Arabidopsis* development.

**NCS1 is required during compromised TDP biosynthesis de novo**. The At5g03555 locus in *Arabidopsis* has been named by two independent groups as *NCS1*[12] and *PLUTO*[13] in accordance with its ability to transport nucleobases (uracil, cytosine) in planta and yeast[12], as well as in *Escherichia coli*[13]. Our attention was drawn to this transporter (using the *NCS1* nomenclature from here) during the course of this study because while there is strong homology at the amino acid level between *NCS1* and

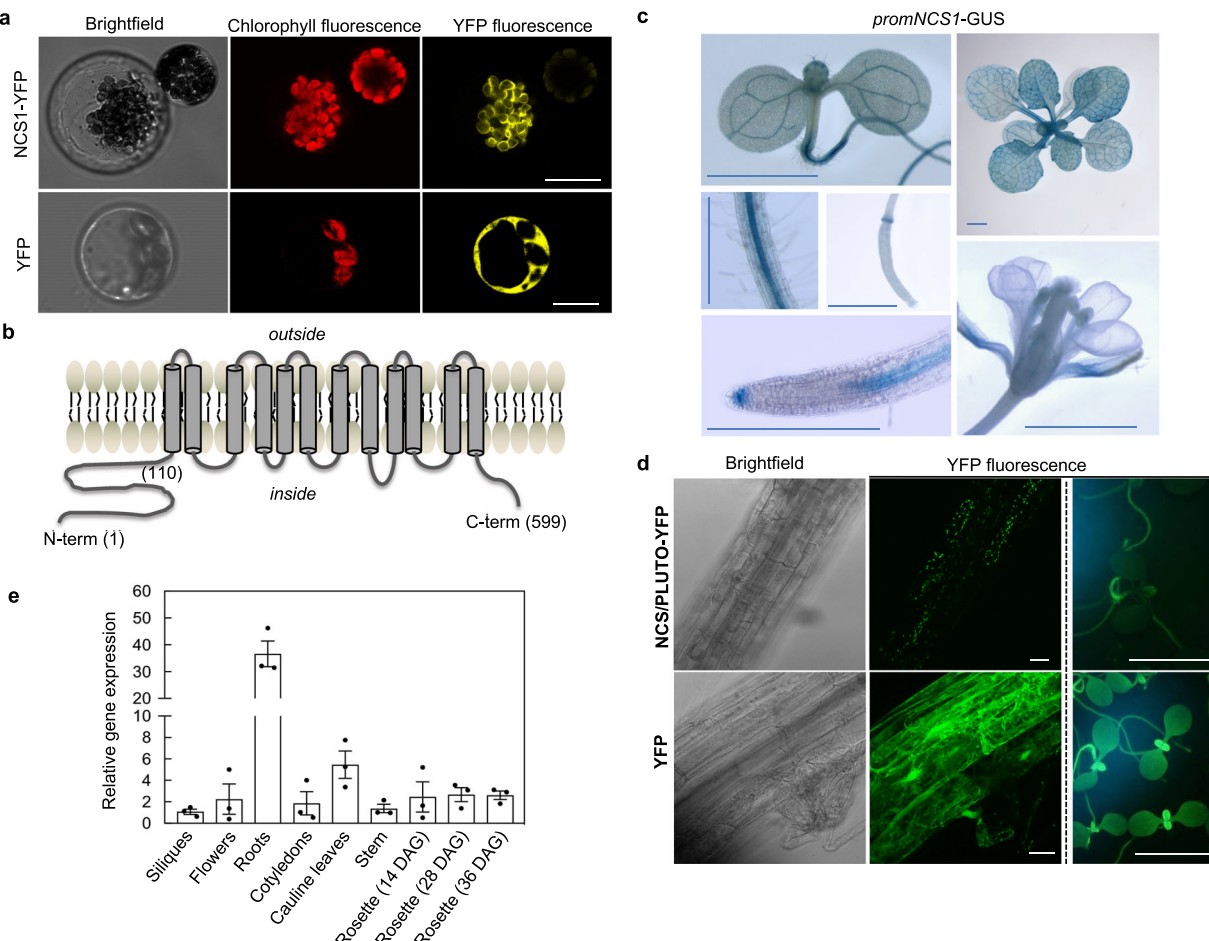

**Fig. 3 The plastid localized NCS1/PLUTO is predominant in root tissue. a** Representative pictures of *Arabidopsis* mesophyll protoplasts expressing the NCS1/PLUTO-YFP fusion protein compared to YFP alone. The scale bar represents 10 μm. **b** Cartoon of the predicted topology of NCS1/PLUTO as determined by TMpred (http://www.ch.embnet.org/software/TMPRED_form.html). The numbers refer to the corresponding amino acid residues. **c** Representative images of histochemical analysis of *Arabidopsis* expressing *GUS* under the control of the promoter of *NCS1/PLUTO*: 7-d-old seedling, differentiation zone of a mature root and a developing silique, 7-d-old seedling root tip, mature rosette and a flower. The scale bars represent 0.1 cm. **d** Representative pictures of tissue expression of NCS1/PLUTO-YFP compared to YFP alone in stable transformants of *Arabidopsis* either by confocal microscopy of roots (columns 1 and 2, the scale bar represents 100 μm) or epifluorescence of young seedlings and roots (column 3, the scale bar represents 0.2 cm). **e** Tissue expression analysis of *NCS1/PLUTO* transcript levels by qPCR. Data of three individual biological replicates is shown with error bars representing SE. Transcript levels are relative to *PDF2* (At1g13320) and each sample was referenced to siliques (set to 1). Plants were either grown on 1/2 MS agar plates or on soil under a 16-h photoperiod (120 μmol photons $m^{-2} s^{-1}$) at 22 °C and 8 h of darkness at 18 °C.

nucleobase permeases in *Saccharomyces cerevisiae*, such as *DAL4*, *FUI1* and *FUR4*, there is also significant homology to *THI7* (30.4% similarity, 16.7% identity), which has been shown to be involved in import of thiamine in *S. cerevisiae*[27,28]. Moreover, homologs of *NCS1* in bacteria cluster with thiamine biosynthesis and salvage genes[14]. Indeed, the latter recent study indicates that *NCS1* may also transport the nonphosphorylated pyrimidine moiety precursor of TDP, hydroxymethylpyrimidine (HMP, Fig. 1). Thus, NCS1 may be a promiscuous transporter but the nature of its operation and specificity is currently undefined. Given the implication of *NCS1* in transport of thiamine molecules, we were prompted to look further at this gene in the context of this study. Firstly, fluorescence microscopy of a translational fusion of NCS1 to YFP confirmed its presence at the chloroplast membrane in *Arabidopsis* protoplasts (Fig. 3a). It should be noted that while the NCS1 is predicted to have 12 transmembrane domains similar to its counterparts in microorganisms, the plant protein exhibits an N-terminal extension (ca. 110 residues) before the first predicted transmembrane domain (Fig. 3b) not seen in its homologs in lower organisms,

which is predicted to represent a plastid transit peptide based on several informatics tools (iPSORT, http://ipsort.hgc.jp; TargetP, http://www.cbs.dtu.dk/services/TargetP/; ChloroP, http://www.cbs.dtu.dk/services/ChloroP/). Indeed, we tested NCS1 for complementation of the yeast knockout mutant CVY4 (MATα *his3Δ1 leu2Δ0 lys2Δ0 ura3Δ0 thi4Δ::his5+ thi7Δ::KanMX4 thi71Δ::LEU2 thi72Δ::LYS2*)[27]. This yeast strain is unable to either biosynthesize or transport thiamine and can only grow upon supplementation with at least 120 μM thiamine[29]. Neither the full-length *NCS1* sequence nor a truncated version missing the N-terminal 97 amino acids (and consequently devoid of the plastid targeting peptide) could restore the yeast *CVY4* strain to prototrophy (Supplementary Fig. 3). This suggests that either the NCS1 protein is not functional in thiamine transport as expressed at least in yeast—perhaps due to inappropriate localization due to the absence of plastids in yeast, or it requires a plant-specific factor.

We also evaluated the expression of *NCS1* at the tissue level by generating plants expressing the β-glucuronidase (*GUS*) reporter gene under control of the region immediately upstream of the

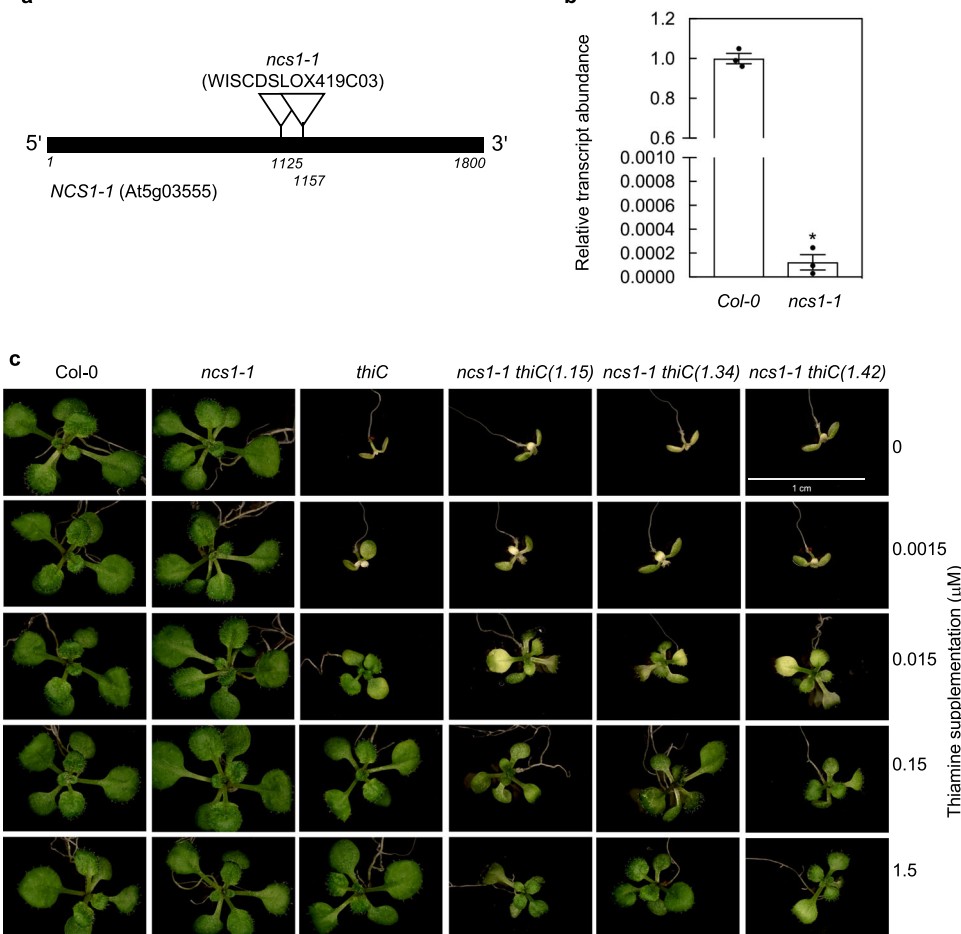

**Fig. 4 NCS1/PLUTO is required for plant development during compromised TDP biosynthesis de novo. a** Gene model of *NCS1/PLUTO* with the single exon depicted as a black bar. The location of the T-DNA insertions in WISCDSLOX419C03 (*ncs1-1*) as depicted were confirmed by genotyping and sequencing. **b** Quantitative analysis of *NCS1/PLUTO* expression in *ncs1-1* relative to wild type (Col-0) from 11-day-old seedlings. Data of three individual biological replicates is shown with error bars representing SE. The asterisk indicates a significant difference as a result of a *t* test ($p \leq 0.01$). Transcript levels are relative to *PDF2* (At1g13320). **c** Representative photographs of 13-day-old *ncs1-1, thiC* and *ncs1-1 thiC* (the numbers refer to three independent crosses) mutant lines in the absence (0) or presence of thiamine supplementation levels as indicated compared to wild type (Col-0). Seedlings were grown on 1/2 MS agar plates under a 16-h photoperiod (120 µmol photons m$^{-2}$ s$^{-1}$) at 22 °C and 8 h of darkness at 18 °C.

transcriptional start site of *NCS1* (*promNCS1-GUS*). In all lines examined, promoter-driven GUS activity, although generally present in leaves and cotyledons, is much more pronounced in the vascular tissue from young and mature leaves and the central cylinder of roots but is weak in flowers and developing siliques as well as developing seeds (Fig. 3c). In general, we observed that the GUS activity was stronger in the root than in the shoot. Interestingly, we frequently observed prominent *promNCS1-GUS* activity in the root apical meristem (Fig. 3c). An examination of roots of *Arabidopsis* lines that stably express *NCS1-YFP*, although under control of the *CaMV 35S* promoter, also showed stronger fluorescence in root tissue compared to green tissue (Fig. 3d). Furthermore, based on the data in Fig. 3a, the most reasonable interpretation of the pattern of dotted fluorescence in these samples is of plastid localization and is in contrast to the diffuse fluorescence pattern typical of a cytosolic localization observed in plants expressing YFP alone (Fig. 3d). Furthermore, a qPCR analysis of *NCS1* detected transcript in all organs examined, but the level was significantly higher in root tissue than in other tissues (Fig. 3e) and reflects publicly available transcriptomic data in the Genevestigator database[26].

To investigate the functional significance of *NCS1* in planta, we examined the T-DNA insertion WISCDSLOX419C03 line[30], which we isolated to homozygosity in this study based on BASTA™ resistance. The line has T-DNA insertions at base pair 1125 and 1157 relative to the start codon in the At5g03555 locus (depicted in Fig. 4a) and has been previously annotated as *ncs1-1*[12]. A qPCR analysis demonstrated that the respective level of transcript in *ncs1-1* was drastically reduced (Fig. 4b). A comparison of *ncs1-1* with wild type (Col-0) did not reveal an incongruent phenotype under standard growth conditions on soil corroborating a previous study[12], or in sterile culture (Fig. 4c). As *ncs1-1* is capable of TDP biosynthesis de novo, a phenotype associated with TDP metabolism, i.e. TDP or HMP transport may not be readily apparent. Consequently, we crossed *ncs1-1* with the T-DNA insertion mutant *thiC* (SAIL_793_H10)[7], which cannot make HMP de novo and develops a chlorotic phenotype 6 days after germination[7]. As shown earlier in this study, supplementation with at least 150 nM of thiamine in the culture medium suffices to restore a wild-type phenotype to the *thiC* mutant but levels below this are limiting for growth. Interestingly, we noted that the double-mutant *ncs1-1 thiC* was not rescued to the same level of

development as *thiC* alone upon thiamine supplementation (Fig. 4c). In particular, seedlings of *ncs1-1 thiC* were smaller and more retarded in development, which was consistent across independent crosses of *ncs1-1* with *thiC* (Fig. 4c). We therefore conclude that *NCS1* is important for the growth of plants particularly when defective in TDP biosynthesis de novo.

**Temporal separation of TDP biosynthesis and transport**. On a daily basis photosynthetic cells need to make dramatic metabolic adjustments during the transition from light to dark to maintain energy supply. As TDP supply needs to be tightly coordinated with the needs of the key enzymes essential for metabolic homeostasis, we were next driven to study the relationship between TDP biosynthesis de novo and transport over the course of the day. Firstly, we searched the diurnal database (http://diurnal.mocklerlab.org/)[31] to assess the diel expression pattern of the known biosynthesis genes, as well as *TPC1/TPC2* and *NCS1* at the transcriptional level. While the data were not available for all genes, both *THIC* and *THI1* the key biosynthesis genes, as well as *TPC1* and *NCS1* indicated rhythmic regulation among three datasets (Supplementary Fig. 4). With this information, our attention was drawn to the tendency of the biosynthesis genes to peak towards, or in, the evening, whereas the transporter transcripts were maximal towards the morning (Supplementary Fig. 4). It is well established that the circadian clock orchestrates the transcript abundance of a large proportion of metabolic genes to synchronize with plant needs[32]. Indeed, *THIC* is known to be transcriptionally regulated by the circadian clock[24]. However, the other genes of TDP metabolism have not been comprehensively investigated. This is important, as circadian control of *THIC* alone may not be solely responsible for coordination of intracellular TDP levels. Thus, to provide more insight into regulation of TDP metabolism, we investigated transcript abundance of all known biosynthesis and transporter genes upon transfer to continuous light after equinoctial entrainment, i.e. circadian conditions. In each case, the significance of oscillation was examined by the algorithms from Biodare2 (https://biodare2.ed.ac.uk/)[33]. While *THIC* oscillation was confirmed, the biosynthesis transcript *THI1* also showed a clear significant oscillation, although the change in amplitude was less pronounced than that of *THIC* (Fig. 5a). Moreover, there was also significant oscillation of the *TPC1* and *NCS1* transcripts, although *TPC2* transcripts were arrhythmic (Fig. 5a). In all cases, any oscillation of transcript expression was abolished in the arrhythmic triple mutant of PSEUDO RESPONSE REGULATOR proteins 5, 7 and 9 (*prr5 prr7 prr9*)[34]. The latter proteins are core components of the circadian clock in plants and their absence renders the internal clock nonfunctional, i.e. arrhythmic[34]. Notably, the tendency for an increase in *CCA1* transcript levels in *prr5 prr7 prr9* is consistent with the known repressor function of the PRR5, 7 and 9 proteins on *CCA1* expression (Fig. 5a). Interestingly, whereas biosynthesis gene transcript abundance peaks in the evening, the transporters were more abundant in the morning (Fig. 5a), reflecting the observations under light-dark cycles. Furthermore, based on published chromatin immunoprecipitation with parallel sequencing (ChIP-seq) data, THIC and THI1 are targeted by CCA1 and LHY[35–37], while NCS1 is targeted by PRR9[38]. Taken together, the data show that transcripts of *THIC*, *THI1*, *TPC1* and *NCS1* are under circadian control and we conclude that TDP biosynthesis and transport are temporally separated at the transcript level.

**Nuclear TDP levels oscillate independent of the clock**. Given the importance of coordinating the supply of the coenzyme TDP with the demands of the cell, the temporal separation of peak abundance of TDP biosynthesis and transporter transcripts prompted us to investigate if free TDP metabolite levels oscillate over a diel cycle. In a previous report on the levels of B₁ vitamers in *Arabidopsis* grown under circadian conditions, thiamine levels could not be detected, TDP was deemed not to oscillate and the TMP vitamer was observed to change in abundance—but only for a single period[24]. Thus, it could not be conclusively deciphered if B₁ vitamer levels demonstrate diel oscillation. To probe this further, we first measured B₁ vitamer levels in shoots of seedlings grown under equinoctial light-dark cycles. In our experiments, all three vitamers could be detected but none showed significant oscillation over 72 h (Fig. 5b).

We next hypothesized that changes in the overall contents of B₁ vitamers may not be observed at the tissue level but rather at the subcellular level to orchestrate intracellular requirements. However, organelle purification is not suitable to assess TDP levels due to the general leakage of soluble metabolites that occurs during the procedure and current methods for nonaqueous fractionation only reveal proportional estimates[39], thereby lacking the required precision. On the other hand, alternative splicing in the 3′-UTR of *THIC* as a function of TDP levels, i.e. the TDP riboswitch, is a natural reporter of cellular-free TDP levels[22,23]. More specifically, the second intron in the 3′-UTR of *THIC* is spliced in the presence of high TDP levels, which we refer to here as *THIC-INTRON SPLICED* (*THIC-IS*), or is retained when TDP levels are low, referred to here as *THIC-INTRON RETAINED* (*THIC-IR*) (Fig. 6a). As the spliceosome machinery is considered to be predominantly localized to the nucleus, we assume that the riboswitch system reports free TDP levels mainly in this organelle. We monitored alternative splicing of *THIC* over either equinoctial light-dark cycles or continuous light (circadian) conditions by qPCR using primer pairs specific to each variant (Fig. 6a). Under equinoctial light-dark cycles, robust rhythms were observed for both *THIC-IR* and *THIC-IS*, although we noted that the peak in abundance of *THIC-IR* was later (beginning of the dark period) than that of *THIC-IS* (8 h into the light) (Fig. 6b). Analysis of the variants under circadian conditions demonstrated a clear robust rhythm for *THIC-IR*, whereas *THIC-IS* was less robust (Fig. 6c). This observation indicated to us that the rhythms observed for alternative splicing of *THIC* are not just a function of the circadian clock, particularly for *THIC-IS*—which directly responds to TDP levels. We therefore examined the *THIC-IR* and *THIC-IS* in the arrhythmic triple-mutant *prr5 prr7 prr9*. Under both equinoctial light-dark cycles and circadian conditions, the rhythm for *THIC-IR* was abolished in *prr5 prr7 prr9* (Fig. 6d, e), implying that the observed oscillation of this variant is predominantly a function of the circadian clock. On the other hand, a robust rhythm was observed for *THIC-IS* under light-dark cycles in the arrhythmic mutant (Fig. 6d), but not under circadian conditions (Fig. 6e). As the presence of the spliced variant is directly related to the abundance of TDP, it therefore appears that there is oscillation of TDP levels independent of circadian clock function but dependent on light-dark cycles. To probe this further and use an independent approach, we generated transgenic *Arabidopsis* lines carrying the firefly *LUCIFERASE* (*LUC*) reporter gene fused to the 3′-UTR of *THIC* with expression under control of the *CaMV 35S* promoter, thereby removing promoter-controlled circadian regulation but retaining TDP responsiveness (Fig. 6f). The abundance of the intron spliced variant of *LUC* (*LUC-IS*), which is proportional to TDP abundance, was monitored under equinoctial light-dark cycles and circadian conditions. A robust rhythm of *LUC-IS* could be observed under equinoctial light-dark cycles, which peaks about 4 h into the photoperiod (Fig. 6g). This oscillation of *LUC-IS* was not observed under circadian conditions (Fig. 6h). Thus, even though the circadian clock is functioning under constant light (free-running) conditions, the *LUC* reporter does not display

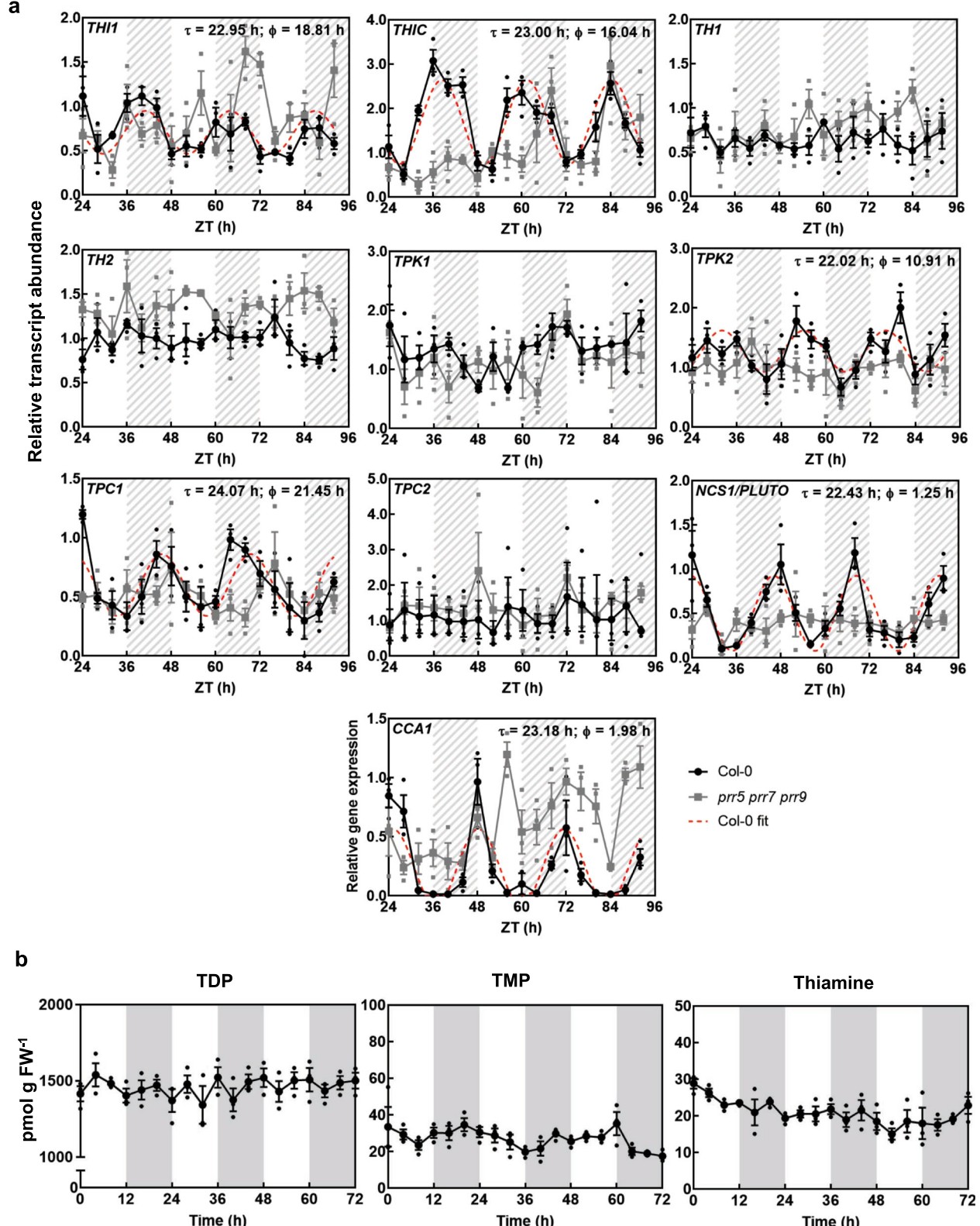

a rhythm. However, oscillation was observed during the first 24 h of constant light as the plant remains under the effect of the last light/dark entrainment cycle during this period (Fig. 6h). These data demonstrate that changes in (mainly nuclear) free TDP levels, as measured by the riboswitch sensor, are a consequence of light-dark cycles. Taken together, we therefore conclude that there is an oscillation of the TDP metabolite at least in the nucleus that is

independent of circadian clock function, and which peaks during the day in light-dark cycles.

## Discussion

It is being increasingly recognized (especially in mammalian cells) that spatial and temporal control of metabolite production may be key to orchestrating specific developmental and physiological

**Fig. 5 Temporal separation of TDP biosynthesis and transport by the circadian clock. a** Transcript abundance of TDP biosynthesis and transport genes by qPCR in continuous light (free-running conditions) in wild type (Col-0, black) compared to the arrhythmic triple-mutant *prr5 prr7 prr9* (gray). The dashed lines correspond to cases where cosine waves could be fitted to the data using the FFT-NLLS method from BioDare2 with the estimated period ($\Phi$) and phase ($\tau$) as indicated. Transcript levels are relative to *UBC21* (At5g25760) and each sample was referenced to wild-type ZT 24. The transcript profile of *CCA1* under the same conditions is shown as a control. Plants were grown in culture on 1/2 MS agar plates and entrained for 13 days in equinoctial conditions (12-h photoperiod with 120 μmol photons m$^{-2}$ s$^{-1}$ and 12 h of darkness at 20 °C) before transfer to constant light after which shoot material was harvested from seedlings ($n = 10$) every 4 h at the ZT times indicated. White and hatched gray areas represent subjective day and night, respectively. Data of three individual biological replicates are shown with error bars representing SE. **b** Thiamine, thiamine monophosphate (TMP) or thiamine diphosphate (TDP) content of wild type (Col-0), as determined by HPLC from shoot material of 14-day-old seedlings grown in culture on 1/2 MS agar plates under equinoctial conditions (12-h photoperiod with 120 μmol photons m$^{-2}$ s$^{-1}$ and 12 h of darkness at 20 °C). Data of three individual biological replicates of pooled material ($n = 20-25$) is shown with error bars representing SE.

outcomes[40]. Metabolite production is inherently dependent on the activity and regulation of the corresponding biosynthesis enzymes. Regulation of enzyme activity is investigated extensively with regard to post-transcriptional and post-translational modifications. However, regulation based on the availability of the corresponding coenzyme, when used, is virtually ignored. As the level of enzyme activity that depends on a coenzyme is directly proportional to its stoichiometry with its corresponding coenzyme, i.e. amount of apoenzyme to holoenzyme, then control of coenzyme levels warrants attention. In this case, we have studied the coenzyme TDP, essential for the activity of numerous key regulatory enzymes within core metabolism. We demonstrate that supply of this coenzyme is not only essential but that its limitation also restricts the abundance of other central metabolites (Fig. 2). In particular, the thiamine dosage response data show that core biochemical pathways are affected and more specifically the level of certain photosynthetic pigments can be correlated with thiamine supply. Often referred to as the "energy vitamin"[41], TDP drives the activity of pyruvate dehydrogenase (glycolysis), pyruvate decarboxylase (glycolysis), α-ketoglutarate dehydrogenase (TCA cycle) and transketolase (oxidative pentose phosphate pathway) in all organisms. Notably, the latter enzyme is also involved in the Calvin cycle, responsible for $CO_2$ fixation in plants. These are all key regulatory enzymes within their respective pathways and thus, the efficiency of the system is inherently dependent on TDP supply. As the final step of TDP biosynthesis takes place in the cytosol, it is presumed that it needs to be transported into the organelles to facilitate catalysis by the endogenous enzymes dependent on it as a coenzyme (Fig. 1). Consequently, control of both TDP biosynthesis and transport is expected to be fundamental for plant survival. While previous studies have focused on individual mutants of the TDP biosynthesis pathway, it is important to recognize that transport of the coenzyme is also important. In particular, we show here that the two facilitators of mitochondrial TDP transport (*TPC1* and *TPC2*) are vital for the plant. On the other hand, *NCS1/PLUTO* appears to be dispensable for plant growth, at least under our laboratory conditions, and a bona fide transporter of TDP into the chloroplast remains to be identified. Nonetheless, it is well established that biosynthesis de novo of TDP is predominantly restricted to green tissue[42]; thus the abundance of *NCS1/PLUTO* in root tissue may be important in this context, e.g. in non-photosynthetic plastids. Indeed, when the plant is impaired in TDP biosynthesis de novo, loss of *NCS1/PLUTO* compromises plant health, retarding growth and development (Fig. 4).

Here we also show that both biosynthesis and transport of TDP are regulated at the transcriptional level by the circadian clock (Fig. 5a). While this is not surprising, it is interesting that there is temporal separation of the peaks in transcript abundance for both processes (Fig. 5a). Timing of delivery to the organelle is important because coenzyme availability in the organelle could help to synchronize the corresponding enzyme activities with the needs of the cell. For example, α-ketoglutarate is transported out of the mitochondrion at night but into it during the day likely influencing the net activity of α-ketoglutarate dehydrogenase (αKGDH) and its requirement for TDP. Limitation of TDP in the mitochondria would influence respiration. Indeed it has very recently been established in yeast that thiamine supplementation shifts metabolism to respiration rather than fermentation[43]. The purpose of circadian control of biosynthesis and transport may be intertwined in the different metabolic flux requirements during the light and dark. In this context, TDP-dependent mitochondrial enzymes easily lose the coenzyme upon isolation, which suggests a weaker binding of TDP by these enzymes and thereby possible benefits of transport[44]. This could be important for an effective regulation of enzyme activity or for a more sensitive detection of TDP biosynthetic needs. Currently, it is not known how the organellar transporters are regulated and how the flux of TDP into the organelles is orchestrated. However, it is likely that the TPCs act as gatekeepers with respect to TDP delivery, and are involved in fine-tuning flux through enzymes dependent on it as a coenzyme. This question, while not addressed in this study, will be challenging in *Arabidopsis* given the observation here that the TPC transporters are essential for viability. It is interesting that only *TPC1* is under control by the circadian clock and may indicate that it is the more important paralog with respect to timing of TDP delivery.

During diel cycles, as part of a division of labor, metabolites are produced and dispatched into cell organelles to mediate growth. On a daily basis, plants have to coordinate metabolic homeostasis, while coping with the light to dark transition and the loss of an energy source during the night (i.e. photosynthesis). The circadian clock anticipates these changes and its importance is well established, as growth is suboptimal if the diel period deviates from a 24-h cycle[45]. Although there is established circadian regulation of transcripts for proteins involved in fluxes of various molecules[46], there is little evidence of oscillations of metabolite movement at the intracellular level, mainly due to difficulties to measure metabolites in the cell compartments[47]. A key finding in this study is that the coenzyme TDP itself displays a diel rhythm and moreover, that it is not dependent on circadian clock function (Fig. 6). We are very tempted to postulate that the ebb and flow of TDP supply and demand over the light and dark phases of a diel cycle drive the observed rhythm. Notably, the observed peak in free TDP levels is during the day (Fig. 6). If the biochemistry of TDP biosynthesis is considered and the fact that the THIC enzyme in particular is dependent on the thioredoxin-ferredoxin system for reducing equivalents[7], which is only active during photosynthesis, then production of TDP during the light period is obvious and likely builds up before its deployment into the organellar powerhouses.

The TDP rhythm observed here is a function of alternative splicing of a 3′-UTR RNA aptamer and thus, we assume predominantly relates to free TDP levels in the nucleus. It is

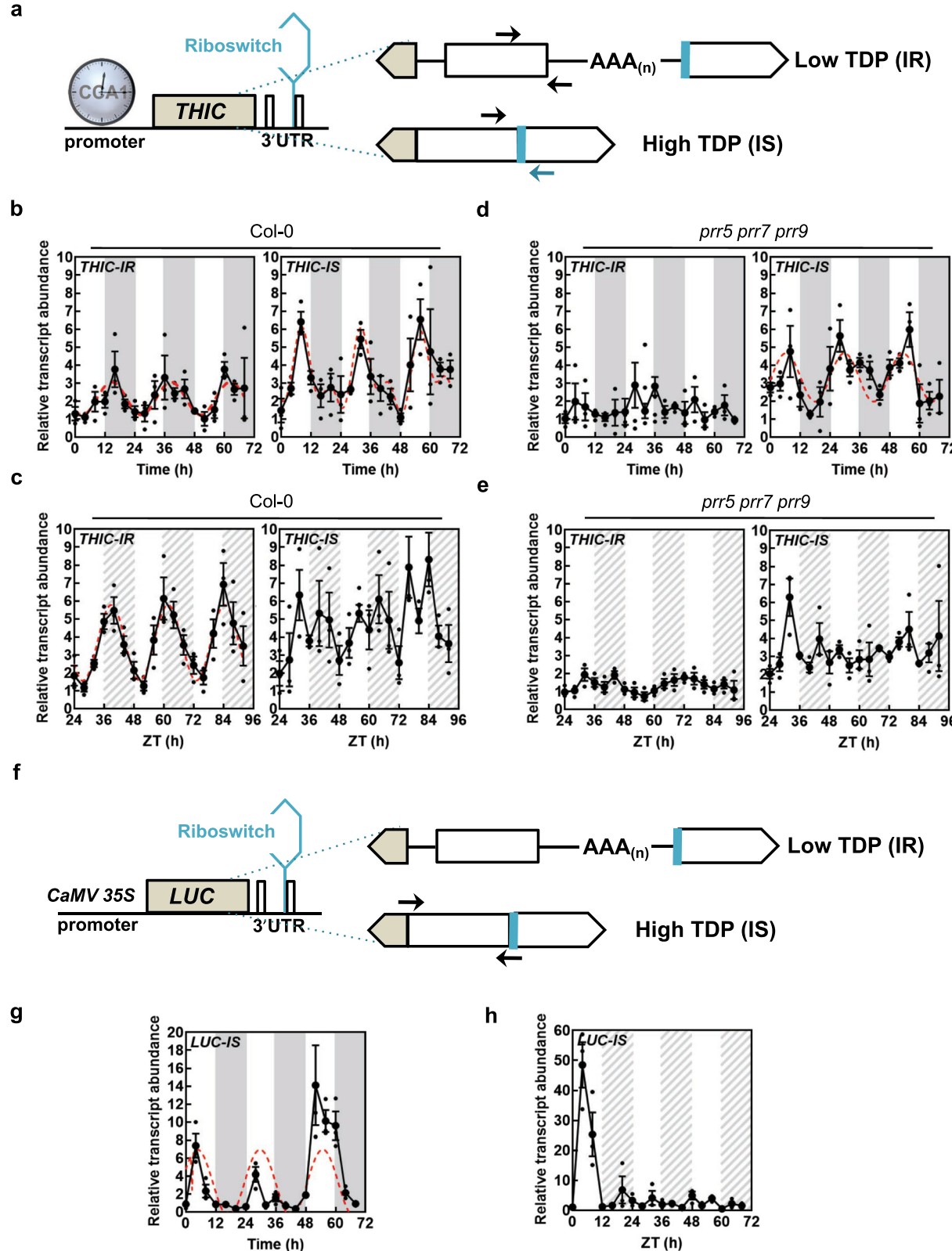

intriguing why an essential coenzyme displays a diel rhythm in the nucleus, a location where presumably there is no apoenzyme that requires it. Why not simply control expression of the corresponding genes through promoter elements? The diel peak of nuclear TDP levels may represent a "relief valve" that serves to switch off functional THIC production directly (via the riboswitch), independent of anticipation by the clock. This thereby ensures a failsafe procedure such that TDP supply is strictly coordinated with demand in the event of circadian system malfunction. That this rhythm is important is supported by the recent observations that firstly disabling riboswitch function has a severe negative impact on plant performance[24] and secondly, that such plants are unable to achieve metabolic homeostasis when suddenly shifted to a different photoperiod length[48].

**Fig. 6 Nuclear TDP levels oscillate in a circadian clock independent manner. a** Gene model of *THIC* emphasizing splice variants of the *3′-UTR* as a function of TDP levels. Exons are depicted as white/beige boxes, introns as lines and the polyadenylation (AAA(n)) tail is as indicated. CCA1 binds in the promoter region of *THIC*. The TDP riboswitch (blue) is at the end of the second intron in the *3′-UTR*. When TDP levels are low the intron is retained (IR), whereas the intron is spliced (IS) when the TDP level is high. The arrows indicate primer-binding sites for monitoring IS and IR levels. **b**, **c** Transcript abundance of IR or IS variants of *THIC* (*THIC-IR* or *THIC-IS*, respectively) by qPCR in equinoctial light/dark cycles (**b**) or continuous light (**c**) in wild type (Col-0). White and dark gray areas represent light and dark in equinoctial conditions, whereas white and hatched gray areas represent subjective day and night in continuous light conditions. Data of three individual biological replicates are shown with error bars representing SE. The red dashed lines correspond to cases where cosine waves could be fitted to the data using the FFT-NLLS method from BioDare2. **d**, **e** As for **c**, **d** but in the arrhythmic triple-mutant *prr5 prr7 prr9*. **f** Gene model of the *LUCIFERASE* (*LUC*) construct expressed under the control of the *CaMV 35S* promoter and with the *THIC 3′-UTR*. Alternative splicing is as for **a** and is dependent on TDP levels in the nucleus. Arrows indicate primer-binding sites in the *3′-UTR* for monitoring IS levels (*LUC-IS*). **g**, **h** Transcript abundance of IS variants of *LUC* (*LUC-IS*) by qPCR in either equinoctial light/dark cycles (**g**) or continuous light (**h**) in wild type. Data of three individual biological replicates are shown with error bars representing SE. The red dashed line corresponds to cosine wave fitting of the data using the FFT-NLLS method from BioDare2. In all cases, plants were grown in culture on 1/2 MS agar plates and entrained for 13 days in equinoctial conditions (12-h photoperiod with 120 μmol photons m$^{-2}$ s$^{-1}$ and 12 h of darkness at 20 °C) and either transferred to constant light or retained in equinoctial conditions. Shoot material was harvested over 3 days at 4-h intervals from seedlings (*n* = 10) at the times indicated. Transcript levels are relative to *UBC21* (At5g25760).

In summary, TDP is biosynthesized in the cytosol and is actively transported into subcellular organelles to be used by enzymes dependent on it as a coenzyme. Changing demands during the light and dark periods of a diel cycle are anticipated by the circadian clock and may necessitate subtle alterations in TDP levels that are fine-tuned in the nucleus at least. Excessive TDP is sensed by the riboswitch. Use of this sole natural metabolite reporter in plants has permitted us to demonstrate that diel conditions are intimately associated with changes in TDP levels. This system can run independently of the circadian clock and serves to maintain TDP homeostasis in line with the needs of the cell and optimize central metabolism[24,48]. Future work will establish the boundaries to this system and how far it can be pushed by altering the sensing of TDP and also by investigating the necessity of this metabolic rhythm and if this is exclusive to the light-dark cycle coincident with metabolic reprogramming in plants and will elaborate on metabolic crosstalk across organelles.

## Methods

**Plant material and growth conditions**. In general, *Arabidopsis* lines were either grown on soil or sterile culture under 12- or 16-h photoperiods (100−150 μmol of photons m$^{-2}$ s$^{-1}$ at 22 °C) as specified, and the corresponding dark period (12 or 8 h at 18 °C) to complete a diel cycle. Seeds set for in vitro culture were surface-sterilized, stratified for 3 days in the dark followed by growth on half-strength Murashige and Skoog medium[49] in 0.55% (w/v) agar (Duchefa) plates (1/2 MS agar plates). Seedlings for global metabolite analyses were grown in sterile culture for 11 days under a 16-h photoperiod (100–150 μmol of photons m$^{-2}$ s$^{-1}$ at 22 °C) and 8-h dark period (at 18 °C) and in the presence or absence of thiamine supplementation as indicated. For diurnal and circadian experiments, seedlings were grown for 13 days in equinoctial conditions (12-h photoperiod with 120 μmol photons m$^{-2}$ s$^{-1}$ and 12 h of darkness at 20 °C) and either kept in these conditions (diurnal experiments) or transferred to constant light (circadian experiments). Shoot material was harvested from seedlings (*n* = 10) every 4 h at the times indicated. For the tissue expression analysis, roots, stems, flowers, siliques, rosette leaves and cauline leaves were collected (*n* = 10) from plants grown on soil under a 16-h photoperiod (100–150 μmol of photons m$^{-2}$ s$^{-1}$ at 22 °C) and 8-h dark period (at 18 °C). Cotyledons were collected at 2 weeks from seedlings grown under the same conditions. In all cases, samples of plant material were immediately frozen in liquid nitrogen and maintained at −80 °C until analysis. The *ncs1*-1 (WISCDSLOX419C03, N854962), *tpc1*-1 (SAIL_130_D09, N806372), *tpc1*-2 (GK-236B06, N422578), *tpc2*-1 (GK-870B10, N483446) and *tpc2*-2 (SAIL_127_G03, N862535) lines were obtained from the European *Arabidopsis* Stock Centre. In all cases, the T-DNA insertion lines could be isolated to homozygosity based on resistance to either BASTA™ (25 μg ml$^{-1}$) or sulfadiazine (7.5 μg ml$^{-1}$) as appropriate, as well as PCR genotyping. Reciprocal crosses of *tpc1*-1 and *tpc2*-1 were performed in an attempt to isolate double-mutant *tpc1 tpc2* lines. The *thiC* line was from an in-house stock described in ref. [7], the triple-mutant *prr5 ppr7 prr9* line was a gift from Kazuki Sato (RIKEN, Japan) and in all cases the Columbia ecotype was used as wild type. Seeds of the crossed lines *thiC ncs1*-1 were selected by PCR analysis as well as the distinctive chlorotic phenotype of *thiC*[7].

**Gene expression analysis by qPCR**. Total RNA was extracted using RNA NucleoSpin Plant kit (Machery-Nagel) following the protocol provided by the company and treated with RNase-free DNase to remove traces of DNA. A total of 1 μg of RNA was reverse transcribed into cDNA using Superscript III reverse transcriptase (Life Technologies) and either oligo(dT)$_{20}$ or oligo(dT)$_{15}$ primers (Life Technologies or Promega, respectively) according to the manufacturer's recommendations. Quantification analyses were performed in 384-well plates using either a 7900HT or QuantStudio5 Fast real-time PCR instrument (Applied Biosystems) by fluorescence-based real-time PCR using Power(Up) SYBR Green master mix (Applied Biosystems) and the following amplification program: 10 min denaturation at 95 °C followed by 40 cycles of 95 °C for 15 s and 60 °C for 1 min. The data were analyzed using the comparative cycle threshold method (2$^{-\Delta CT}$) normalized to the reference gene *PDF2* (At1g13320) or *UBC21* (At5g25760), as indicated. Primers used are listed in Supplementary Table 2. Each experiment was performed with at least three biological and three technical replicates.

**Transgenic plant lines and fluorescence microscopy**. The NCS1-YFP construct was generated by amplifying the full-length sequence of *NCS1* from *Arabidopsis* from cDNA of 10-day-old seedlings and cloned into the pENTR-D/TOPO vector using the pENTR TOPO cloning kit (Life Technologies) according to the manufacturer's instructions and subsequently cloned into Gateway® destination vector pB7YWG2[50] by an LR reaction using LR clonase enzyme mix II (Life Technologies) to be expressed as a fusion protein with YFP at the C terminus (NCS1-YFP). Primers used (*NCS1-YFP*) are listed in Supplementary Table 2. Transient expression in *Arabidopsis* protoplasts with the empty vector as control was carried out as described in ref. [51]. For the generation of transgenic lines the *NCS1-YFP* construct and the empty vector as a control were introduced into *Agrobacterium tumefaciens* strain C58 and used to transform wild-type (Col-0) *Arabidopsis* plants by the floral dip method[52]. Transformants were selected by resistance to BASTA™, allowed to self-fertilize and homozygous lines were selected from the T3 generation according to their segregation ratio for BASTA™ resistance. Eight-day-old transgenic seedlings were mounted in water in a flow chamber and subjected to fluorescence analysis (cotyledon, hypocotyl, and root tissue) either by stereomicroscopy or on an SP2 confocal laser-scanning microscope (Leica), as described previously[53]. The *CaMV 35S:LUC-THIC3′UTR* was constructed from plasmids *Topo-TA-THI-C3′UTR* and *pBinAR-FLUC*[22] kindly provided by Andreas Wachter (ZMBP, Germany) by digestion with *Sal*1, which releases the *THIC3′/UTR* from *Topo-TA-THIC3′UTR* and facilitates its cloning into *pBinAR-FLUC*, thereby generating *CaMV 35S:LUC-THIC3′UTR*. Transgenic *Arabidopsis* carrying this reporter construct was generated by transforming wild type (Col-0) using the floral dip method[52]. Transgenic lines were selected by measuring the luciferase activity of 10-day-old seedlings grown on 1/2 MS agar plates and supplied with 5 mM D-luciferin 24 h prior to imaging using a CDD ORCA2 C4742-98 digital camera (Hamamatsu). Bioluminescent plants were allowed to self-fertilize and homozygous lines were obtained from the T3 generation and responsiveness to thiamine was validated by spraying with 1.5 μM thiamine 24 h prior to imaging as above.

**Generation of GUS lines and staining for GUS activity**. A 519 bp region upstream of the start codon of *NCS1* was amplified by PCR from genomic DNA and inserted into pENTR/D-TOPO (Life Technologies) followed by recombination into the destination vector pMDC162[54]. The corresponding *promNCS1-GUS* construct was used for *Agrobacterium tumefaciens* transformation of *Arabidopsis* (Col-0) by the floral dip method[52]. Transformants were selected on hygromycin (50 mg l$^{-1}$) followed by segregation analysis to isolate lines with a single insertion and was confirmed by PCR analysis. Between 10 and 25 lines were analyzed. Seedlings or tissues were collected in glass vials filled with ice-cold 90% acetone and incubated for 20 min at room temperature. For histochemical localization of GUS activity, the plant material was washed three times with staining buffer (10 mM NaH$_2$PO$_4$/Na$_2$HPO$_4$ buffer pH 7.0, containing 0.5 mM K$_3$[Fe(CN)$_6$], 0.5 mM

$K_4[Fe(CN)_6]$, 0.1% Triton X-100, 10 mM EDTA-Na$_2$) and afterwards infiltrated for 25–90 min (depending on the sample) with 5-bromo-4-chloro-3-indolyl-β-glucuronic acid in staining buffer (0.1 mg ml$^{-1}$). Excess staining solution and chlorophyll was removed from the samples by rinsing several times with 70% ethanol. Images were captured using a Leica MZ16 stereomicroscope equipped with an Infinity 2 digital camera (Leica Microsystems).

**Yeast experiments**. *Saccharomyces cerevisiae* knockout mutant CVY4 (MATα *his3Δ1 leu2Δ0 lys2Δ0 ura3Δ0 thi7Δ::kanMX4 thi71Δ::LEU2 thi72Δ::LYS2 thi4Δ:: his5$^+$*) deficient in the biosynthesis and transport of vitamin B$_1$[27] and that only grows in the presence of high concentrations of thiamine (≥120 µm) was used. *S. cerevisiae* BY4742 (MATα *his3Δ1 leu2Δ0 lys2Δ0 ura3Δ0*) was used as the wild-type strain. Synthetic dextrose medium lacking vitamin B$_1$ and uracil (Formedium) and containing 2% (w/v) glucose was used as the basic growth medium and supplemented with the indicated concentrations of thiamine. The cDNA corresponding to *Arabidopsis NCS1* was amplified using the primer pairs listed in Supplementary Table 2 and cloned by homologous recombination into *pGREG506-TEF* according to ref. [55] to generate *pGREG506-TEF-NCS1* (AtNCS1 FL). The AtNCS1 Δ1–97 construct was prepared in a similar manner using the primers listed in Supplementary Table 2. The strain harboring *pGREG506-TEF-PUT3* (annotated as AtPUT3 in Supplementary Fig. 3) was as described in ref. [29].

**Nontargeted metabolomic analysis**. Whole seedlings of 11-day-old *Arabidopsis* wild type (Col-0) or the *thiC* mutant (SAIL_793_H10[7]) grown in the absence or presence of thiamine supplementation (0–1.5 µM) were used. Seeds were surface-sterilized with ethanol and sown on culture plates containing Murashige and Skoog (MS) medium[49] and 0.55% agar (Duchefa) for growth under long day conditions (60% relative humidity, 100–150 µmol photons m$^2$ s$^{-1}$ and 22 °C for 16 h followed by 8 h of darkness at 18 °C) and ambient CO$_2$. For metabolite extraction, fresh plant material (10–50 mg) was flash frozen and homogenized in 75% ethanol containing pre-warmed (80 °C) 10 mM ammonium carbonate, pH 7.5 and shaken for 4 min at 80 °C followed by centrifugation. The supernatants were analyzed by nontargeted metabolomics using flow injection analysis—time-of-flight mass spectrometry on an Agilent 6520 instrument[56]. After data curation and filtering, 372 ions with distinct mass-to-charge could be associated to one or more deprotonated metabolites of *Arabidopsis* listed by the Kyoto Encyclopedia of Genes and Genomes (https://www.genome.jp/kegg/). For all detected metabolites, we performed a differential analysis comparing thiamine supplemented samples with "0" controls. Out of all 372 ions, 24 were found to be significantly changed in at least one condition (|log2(fold-change)| > 0.5 and adj. *p* value < 0.01).

**B$_1$ vitamer determination by HPLC**. Determination of B$_1$ vitamer content was essentially carried out as described by ref. [29]. Unless indicated otherwise, two-week-old seedlings (50 mg) were homogenized in 100 µl 1% TCA and incubated at room temperature for 30 min with constant shaking. Proteins were precipitated by centrifugation and the supernatant was neutralized with 1 M Tris-HCl, pH 9.0, containing 50 mM MgCl$_2$ (10% v/v). Thiamine and its phosphorylated forms were then chemically converted to their corresponding thiochromes by alkaline oxidation with potassium ferricyanide (K$_3$Fe(CN)$_6$) according to ref. [57] with minor modifications: Seven microliters of a freshly made solution of 46 mM K$_3$Fe(CN)$_6$ (in 23% NaOH (w/v)) was added to 50 µl of extracts and then incubated in complete darkness for 10 min. Twelve microliters of 1 M NaOH and 25 µl of methanol were subsequently added. The extracts were clarified by centrifugation prior to injection into the HPLC. Quantification was carried out using the linear range of a standard curve constructed with known amounts of B$_1$ vitamers using an Agilent 1200 series HPLC equipped with a binary pump, an autosampler and a fluorometer (excitation 370 nm, emission 430 nm; Agilent Technologies). Separation was achieved at 35 °C using a COSMOSIL π-NAP column (150 mm × 2.0 mm I.D., Nacalai Tesque Inc., Kyoto). Mobile phase A was 50 mM potassium phosphate buffer, pH 7.2 and mobile phase B was methanol. Gradient steps were programmed as follows: 11% B ramped to 32% in 9.5 min, 32% B ramped to 60% in 3.5 min, 60% B ramped to 100% in 0.5 min and held at 100% B for 2 min, returned to initial conditions during 0.5 min, and equilibrated for 6 min. Analytes were eluted at a flow rate of 0.5 ml/min. Injection volume was 10 µl.

**Circadian rhythm and statistical analyses**. Time-series data were obtained from three biological replicates and each measured by qPCR in technical triplication. The mean values of each time series were analyzed using three different period analysis algorithms FFT-NLLS, MFourfit and MESA from Biodare2 (https://biodare2.ed.ac.uk/)[33] to identify time series that showed a circadian rhythm. Data were linearly detrended and a data analysis window of at least three consecutive cycles excluding the first 24 h of continuous light was used for period estimates. Period and phase estimates and cosine curve fitting were based on values obtained from the FFT-NLLS algorithm. Statistical analyses, *t* tests, one-way ANOVA and two-way ANOVA, were performed in GraphPad Prism v7.

**Data availability**

The source data underlying plots shown in the main figures of this study are available in Supplementary Data 1.

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

## Acknowledgements

We gratefully acknowledge the Swiss National Science Foundation (grants 31003A-141117/1 and 31003A_162555/1 to T.B.F.), as well as the University of Geneva for supporting this work. We thank Michael Moulin for preliminary investigations during the course of this study, Mireille de Meyer-Fague (University of Geneva) for assistance with *Arabidopsis* lines, Kazuki Sato (RIKEN, Japan) for the triple-mutant *prr5 ppr7 prr9* line and Andreas Wachter (ZMBP, Germany) for supplying plasmids *Topo-TA-THIC3′ UTR* and *pBinAR-FLUC*.

## Author contributions

T.B.F. conceived the study; Z.B.N., and C.T. performed all circadian and diurnal analyses; L.P. and N.Z. performed and analyzed the metabolomics; A.G. and E.G.-P. performed transport line analysis and microscopy; I.D. assisted with all plant work and analyzed data; M.H. and C.R. performed and analyzed B₁ vitamer analysis; T.B.F. wrote the article with assistance from Z.B.N. and C.T. All authors read and approved the manuscript before submission.

## Competing interests

The authors declare no competing interests.
