## [Peer Review File · Communications Biology]

Reviewers' comments:

Reviewer #1 (Remarks to the Author):

This manuscript reports on the diurnal oscillation of the coenzyme thiamin diphosphate (TDP) in the nucleus independently of the circadian clock. To prove this, the authors indirectly measured TDP levels in the nucleus of wild-type and circadian clock Arabidopsis mutants by measuring transcript variants of the biosynthesis gene THIC that are under the control of TDP levels via a unique riboswitch TDP sensor. The authors also show the importance of fine regulation of TDP homeostasis on plant's fitness by controlling the whole plant primary metabolism. They also used Arabidopsis mutants to demonstrate in planta that the mitochondrial TDP transporter TPC is essential for plant's life, and that the plastidial NSC1 (or PLUTO) transporter is important in the absence of de novo TDP biosynthesis. The manuscript is well written, the methods are scientifically sound, and the conclusions are supported by the results. I only have minor comments:

Supplementary Fig. 2e: add legends of squares (black = TPC1; white = TPC2)

The authors should explain why they chose the *tpc1-1* mutant instead of *tpc1-2* mutant in their cross with *tpc2-2* mutant. *tpc1-2* has an insertion in an exon while *tpc1-1* has an insertion in an intron. Insertion in an exon is more likely to produce a complete knock out.

Figure 4b: the figure legend mentions that error bars represent SE, but there is no error bar in the graph.

Line 212: "thirteen days after germination" while Fig 4c indicates day 11. Which one is correct?

Line 277-279: "and under circadian conditions" – I honestly do not see any rhythm on this graph, except the peak within the first 36 hours.

Reviewer #2 (Remarks to the Author):

This is careful work that investigates a set of novel questions relating to vitamin cofactor metabolism in plants. The authors provide important new information demonstrating the importance of B1 vitamin metabolism. They also convincingly demonstrate circadian regulation of the levels of transcripts of the components of this pathway. The study makes technically-demanding measurements of B1 vitamins within timecourse experiments, making this probably one of the first studies to directly investigate the clock control of plant enzyme cofactors. Using an elegant riboswitch reporter, the authors investigate daily changes in some of these molecules at a subcellular scale, which is an interesting finding that also speaks to the broader question of the involvement of circadian regulation in the subcellular compartmentation of metabolism (that broader issue could be emphasized more fully to add impact to the manuscript). There are some places in the article where the level of replication is inadequate to support the conclusions that are reached. The work raises some interesting questions for the future about, for example, the mechanisms of daily regulation of the metabolite rhythms. Could it be driven by photoreceptors or a plastid-based process, for example?

Major points

1. At times, the manuscript appears to be two distinct papers that have been combined. Why did authors decide to focus on clock control rather than other responses, such as photosynthetic regulation or stress tolerance? The paper would work better as a single entity if the authors can make provide a more robust justification for why they chose to focus upon daily and circadian regulation within the second part of their article.

2. I am confused about the authors' thinking about the proposed plant NCS1 homolog. The authors say that the bacterial ortholog of NCS1 might be involved in HMP transport but in the model in Fig. 1, in plants it appears that it is TMP that is transported across the chloroplast envelope instead. Could the authors add a sentence about the degree of specificity possessed by such a transport protein? Could the transporter have promiscuity for these different related metabolites?

3. How do they know [TDP] oscillation is only nuclear? For a non-specialist reader it would be good to have a simple explanation for why the riboswitch system is detecting only nuclear processes.

4. It might make more sense to combine Supplemental Fig. 3 into Fig. 1. It seems that this supplemental figure presents a very similar data type as Fig. 1a and that the two would fit nicely alongside.

5. I do not think the level of replication in Fig. 4c is adequate for any conclusions to be reached from these images. How do we know that the authors are not cherry-picking seedling images that support their hypothesis? It would be much better to quantify some parameters, e.g. cotyledon size or chlorophyll content to provide a proper replicated and quantitative measure. This does not appear to be a well-executed experiment that can be interpreted robustly.

6a. I am confused about the interpretation of the data in Fig. 6b-e in relation to the data in Fig. 5a. What form of the THIC transcript was quantified in Fig. 5a and how does this compare with the two splice forms that were monitored in Fig. 6? It would be really helpful to have a diagram that illustrates the primer design that was used to detect these 2 or 3 classes of the transcript. Given that the THIC transcript has a rhythm in Fig. 5a, could this alternatively indicate that the rhythms in Fig. 6b are due to a rhythm in the strength of the promoter that is regulating THIC-IR and THIC-IS? Did the authors make or try a regular promoter-luciferase reporter for THIC? If the data in Fig. 6 discount this possibility for THIC-IR and THIC-IS (perhaps the different phases of the two splice forms?), it would be helpful for the authors to provide a robust explanation in the text.

6b. These data appear to be over-interpreted in places. The authors state, "On the other hand, a robust rhythm was observed for THIC-IS ... under circadian conditions albeit less pronounced". This is simply not true. THIC-IS does not appear to be rhythmic in the triple mutant under constant conditions. There is no quantitative analysis to support this conclusion and the pattern of peaks in the data do not support this interpretation. The authors appear to use this to reach the conclusion, "it therefore appears that there is oscillation of TDP levels independent of circadian clock function." This statement is true based on the data collected in the triple mutant under diel conditions- and well supported by the subsequent luciferase experiments- but cannot be concluded from the data from the triple mutant under constant conditions.

Minor points/typos/editorial

General: It would be preferable for the authors to use "diel" rather than "diurnal" when they refer to a process that occurs over a day/night cycle. This is inconsistent within the manuscript. "Diurnal" also means the converse of "nocturnal" so is ambiguous.

General: For consistency, it is recommended to be careful about interchanging "gene expression" and "transcript levels". "Transcript levels" is probably more accurate when the data show transcript levels.

General: The authors say "data not shown" at several points in the manuscript. Why have they excluded these data? If they are negative results (e.g. with NCS1 in yeast), why not include them because they are informative?

General: A number of the figures lack scale bars and, depending on journal policies, these might

be needed. For example, Supplemental Fig. 3 might need scale bars for a valid comparison of seedling size.

General: It would help readers to have referencing of the figures within the Discussion text.

General: I find the use of the phrase "clock autonomous metabolic oscillations" to be obscure. Do the authors mean oscillations of metabolic processes that occur independently from the known circadian oscillator, and/or remain rhythmic in the absence of a functional transcription-translation circadian oscillator? If so, I do not think the authors have identified "clock autonomous metabolic oscillations" because the oscillations they have measured appear to be driven by the light/dark cycle.

Fig. 2a; it would be helpful if the authors could provide some quantification of a factor within their images to provide readers with a sense of the quantitative differences and variation in individual seedlings of this response. Could they extract a parameter (such as mean visible plant area per seedling?) for each thiamine concentration? Perhaps the wild type is a bit bigger at 1.5 μ M thiamine, but it's hard to tell without some quantification.

Fig. S2d; given that metabolite concentrations can be variable, it would be sensible for the authors do perform some statistical analysis on these data. For example, is [TDP] lower in *tpc1-1* than the wild type? Alternatively, it could be informative to overlay a dotplot or swarm plot.

Fig. S2e; this seems to be a confusing presentation of the data. Would it not be clearer to divide the graph into two parts, one comparing TPC1 across the developmental stages and anatomy, and one beside this for TPC2?

Fig. 4b; the authors state in the figure legend, "The data are the average of three 726 independent biological replicates with error bars representing SE" yet no such error bars are present on the graph and there is no explanation of any statistical analysis that led to the starring on the plot.

Fig. 5a; why is the transcript level of CCA1 so variable in the triple mutant? Is this expected? This should at least be commented upon briefly within the Results section.

Line 97-98; "in order to demonstrate that" -> "in order to test whether" [the former suggests a lack of scientific equipoise on behalf of the authors].

Line 110-111; does this statement need a caveat that the concentration in certain subcellular compartments might be greater or lower than this approximate threshold of 1500 pmol g⁻¹ FW?

Line 122; when discussing these transport proteins, it is important to explain in the text that At3g21390 is distinct from the two-pore channel TPC1 transport protein (At4g03560). I make this comment particularly because At3g21390 is not yet annotated as TPC1 within the TAIR/Araport databases, whereas the TPC1 Ca²⁺ channel is already annotated. At least with Arabidopsis, At3g21390 should possibly be annotated with a distinct name from TPC1. A sentence of clarification would rule out any potential ambiguity by the readership.

Line 142; the authors should provide very brief justification of why they chose specific alleles of each gene for their crosses.

Line 147; please can the authors expand upon, "a sole embryo based problem"? What does this mean? Could the result suggest a male- or female-specific germline effect?

Line 171; what is revealed by informatic tools about the putative transit peptide? Such an analysis might add confidence to this interpretation. If it appears that the transit peptide is, on probability,

a chloroplast transit peptide then could the results with the yeast complementation experiment be simply interpreted in terms of the fact that yeast doesn't have chloroplasts and therefore the metabolic compartmentation is different in yeast?

Line 179 onwards; no information is provided here or in the methods about the size of the promoter fragment used for the GUS assay. This is essential for reproducibility of the work.

Line 179; the claim about dotted GFP signal being consistent with plastid localization is over-interpretation in the absence of a known plastid marker. It would be better to say that based on the data in Fig. 3a, the most reasonable interpretation is that this is indicative of plastid localization.

Line 263; there is no reason "free" needs to be in italic.

Line 257; "changes in the overall contents..." of exactly what?

Line 302; "While regulation of..." this sentence seems garbled or over-edited. Perhaps divide it into two sentences?

Line 331; "Here we also show that both biosynthesis and transport of TDP are regulated at transcriptional level by the circadian clock." – this might be over-interpretation because the authors do not have information about the activity of the enzymes encoded by these genes. A slightly different form of words would be more parsimonious.

Reviewer #3 (Remarks to the Author):

In this paper by Noordally et al, "The coenzyme thiamine diphosphate displays a nuclear rhythm independent of the circadian clock," the authors explore the role of transporters in thiamine mobility in planta and the effect of daily light cycles on the post-translational regulation. The authors report on the changes in metabolite levels in wild type and thiC mutants in response to supplemental thiamine. The authors report on the lethality of tpc1/tpc2 double mutants, and an interaction of ncs1 with thiC during recovery with supplemental thiamine. The authors also report on the subcellular and tissue specific expression of NCS1. The authors also report on the expression of multiple thiamine metabolism genes and the TPCs where they extrapolate that transport maybe temporally separated from circadian peak, however, changes in TPC1 protein levels are not shown. Finally, the authors explore diurnal vs. circadian peaks for the ribo-switch mediated splicing and observe a diurnal peak of splicing as opposed to the nocturnal peak of expression. This paper is a good start, but incomplete story. Many processes can be rhythmic without clock control, and metabolism is no exception. The major finding is a single panel that splicing has a daily rhythm that is presumably light responsive. However, the authors do not explore this result further. The authors have to show much more for this to be of interest to a diverse audience.

1. A main thrust of this paper is the report that proper post-transcriptional control of a single transcript in the thiamine is driven by the environment, not the clock. Although many forward the idea the metabolic rhythms are influenced by the clock, this idea is far from "the central dogma is that metabolic rhythmicity is a clock driven process" as described in the abstract. This is an overstatement and should be removed from the abstract. Most processes in plants are influenced by the environment (light/dark cycles, temp cycles, rainfall cycles, humidity cycles) and are rhythmic, but independent of the clock, including metabolism. The observation of post-translational regulation with a daily, non-circadian clock driven oscillation is not striking. The authors should reconsider the abstract.

2. The results on NCS1 are very preliminary and not quantified. The authors should quantify the thiamine rescue of the *ncs1 thiC* double mutant (FW biomass, or other measurement) compared to *thiC* alone. Qualitative reports based on single picture is insufficient. Also, the authors should complement mutant line to show that the deficiency in growth (if significant) is due to loss of NCS1.

3. The authors may want to consider testing if NCS1 is a TPP transporter by targeting it to the mitochondria to see if it can complement a *tpc1/tcp2* double mutant. Similarly, the authors could also show that TPP transport is the defect in the *ncs1* mutant by targeting a *tpc* to the chloroplast membrane. These experiments could illustrate that the transporters are indeed moving similar metabolites across membranes.

4. The authors forward the idea biosynthesis and transport are temporally segregated processes, however, the authors old describe the transcription of the genes, and do not show that either biosynthesis or transport are changing over the day. In fact, Figure 5B shows that major products of the thiamine pathway are constant over a 24 hour day.

5. Although the authors forward the idea that there is a daily rhythm of TPP levels as seen from the splicing of the ribo-switch as part of a luciferase reporter construct (Fig 6B), the daily levels of TPP do not change (Figure 5B). How are TPP levels in the nucleus connected to the lack of daily rhythms in the metabolite itself. This is a big disconnect. One possibility is that alternative splicing factors are regulated by light, as is being reported (<https://doi.org/10.1073/pnas.1407147112>, <https://doi.org/10.1093/pcp/pcy089>). It could be that TPP levels are constant (as seen in figure 5B) but the coincidence with light triggers intron retention vs. splicing.

6. The authors show primers and experimental design relative LUC-IR transcripts, but presumably no products are observed since they would require splicing to bind. The authors should design primers similar to those for testing THIC-IR to determine how does the expression of LUC-IR to LUC-IS compare. The authors should quantify the abundance of the spliced vs. un-spliced versions to each other. Also, does the LUCIFERASE reporter (by bioluminescence, as the authors report using to identify the transformed lines in the materials and methods) reflect the IR or IS confirmation in diurnal conditions?

7. Does THIC-IS show an ultradian rhythm in LL (Figure 6c)? It seems that there is a double peak, one during the day following the diurnal peak and another at night when the unspliced transcript peaks. One might be observing the intersection between a circadian-regulated daily peak of transcript expression and post-translational splicing. Higher time resolution might resolve this (2 hr resolution).

Minor criticism:

Line 24: "we then took advantage of the sole riboswitch metabolite sensor in plants..." Please include "identified so far" or "sole reported" as the search for riboswitches has not been exhaustively performed.

Fig 2c: bottom legend for color chart should read ≤ -2

Reviewer #4 (Remarks to the Author):

Review on "The coenzyme thiamine diphosphate displays a nuclear rhythm independent of the circadian clock" by Noordally et al.

Noordally et al. explored in *Arabidopsis* metabolic oscillations of the thiamine diphosphate (TDP), the form of vitamin B1 acting as a coenzyme and that plays important role in carbon metabolism. The authors showed that genes involved in TDP biosynthesis and transport are transcriptionally regulated by the circadian clock and that these two mechanisms are temporally separated at transcript level. Finally, they suggested that oscillations of TDP detected in the nucleus are independent of the clock function.

Noordally et al. performed an interesting work on clock autonomous metabolic oscillations, an original field that is still poorly understood in plants. I appreciated the originality of this work, the clarity of the figures and the interesting discussion. However, some conclusions in the result section are too direct and should be tone down, or additional data should be provided to support these conclusions. I also have a few comments/suggestions to improve the figures. Particularly, scale bars are missing on all pictures and should be added.

Major points:

Introduction:

1. I am not sure it is necessary to describe the mechanisms in the figure legend (Figure 1), but I am ok with that. However, the role of CCA1 in the regulatory processes should be explained in the introduction, not only in the legend.

Results:

2. I.100: "Supplementation with thiamine had little effect on wild type growth (Fig. 2a). However, the growth of *thiC* beyond the cotyledon stage was dependent on the level of thiamine supplementation (Fig. 2a)."

and I.110: "it appears that ca. 1500 pmol g⁻¹ FW is sufficient to sustain growth at wild type levels (Fig. 2b)"

I agree with the authors that *thiC* growth depends on the level of thiamine. However, based on the plant pictures, it seems like the 1.5 concentration do not completely rescue the phenotype observed for *Col-0*, while levels of TDP seem to be similar in the two genotypes under this thiamine concentration. This result needs phenotypic data to confirm the conclusion, such as number of leaves per plant, leaf surface area, fresh mass, dry mass, etc., for each thiamine concentration. Otherwise, the conclusion should be significantly tone down and the authors should discuss the fact that high TDP levels in plants may not be sufficient to completely rescue the *thiC* phenotype. In addition, statistical results should be added to confirm or not that TDP levels and plant phenotypic variables are similar between the two genotypes under 1.5 μ M of thiamine.

3. I.132: "However, no morphological defects could be deciphered compared to the wild type (*Col-0*) grown under the same conditions, suggesting functional redundancy of TPC1 and TPC2 (Supplementary Fig. 2b)."

Unless it is known that *tpc1* and *tpc2* mutants alter plant phenotype, I would remove the second part of the sentence in the absence of a double mutant phenotype. To me, this result does not necessarily suggest a functional redundancy between the two genes. Moreover, the authors demonstrated that TPC1 is cycling under constant light, while TPC2 does not exhibit circadian oscillations (Fig 5a), suggesting that these two genes are not similarly regulated.

4. Scale bars should be added in:

- Fig 2a: for both plant pictures on the left and plate pictures
- Fig 3a, c and d
- Fig 4c

5. Based on published Chip-Seq data, THIC and THI1 are both targeted by CCA1 and LHY, and NCS1 is targeted by PRR9 (see Nagel et al, 2015; Kamioka et al, 2016; Adams et al, 2018; Liu et al, 2016). This information should be added in the text and would add strength to the authors' conclusion on l. 244.

6. l. 294-295. "...we therefore conclude that there is an oscillation of the TDP metabolite in the nucleus that is independent of circadian clock function"

I understand and I agree with this interpretation. However, I am surprised that the authors did not discuss the role of CCA1 in this paragraph, while this gene is part of the core clock machinery and is the main regulator of THIC expression. Also, I think that expression profiles of CCA1 in both LD and LL conditions and in both Col-0 and prr5-prr7-prr9 triple mutant should be added to this figure 6. In prr mutants, oscillations of CCA1 are disturbed, but the gene is still expressed (for example, Salome et al, 2010, The Plant Cell, Fig 2A). This probably explains why THIC-IS transcript is still produced in the triple mutant under constant light (Fig 6e), but this transcript (THIC-IS) is not cycling probably because TDP is not cycling under constant light, as described by the authors. So, I think adding CCA1 expression profiles would add strength to the conclusion and would help the reader to understand this interpretation.

Again, the role of CCA1 should be discussed more in the manuscript.

Minor points:

1. Arrow color code should be explained in the legend of Figure 1. I guess that the blue arrow represents the "negative feedback regulation on TDP biosynthesis de novo" caused by the "binding of TDP to the THIC riboswitch", as mentioned in the introduction (l. 74-76). This should be clarified in the legend.

2. Fig 2b: More numbers on the x-axis should be added.

3. Fig 2c: Please indicate "thiamine supplementation" as a y-axis title and add numbers on the y-axis, remove text at the top of the plot and add a legend for the dot size.

4. Supplementary Fig 1: This figure is highly informative. Maybe I missed it, but I think this data should also be provided as a Supplementary Table, with Pvalues and log2FC. Also, a heatmap of the log2FC values for all significant metabolites (one column for a and one for b) would allow us to easily compare metabolite responses in the two genotypes. This could be Supp Fig 1c.

5. Supplementary Table 1: Please check the table legend "(P > 0.05 ; *P > 0.05)".

6. l.212: "...more apparent thirteen days after germination": 13 days or 11 as indicated on Fig 4c? Please clarify.

7. From DIURNAL, data obtained in LL condition are available and could be plotted in Supplementary Fig 4 as well.

8. Fig 5a: A legend for the two genotypes would facilitate the understanding without the need to read the figure legend.

9. Fig 5a: This panel could be expanded a little bit if you do not repeat the axis titles for each individual plot

10. Fig 5b: If you do not repeat the axis titles and if you indicate the measured compounds at the top of each individual plot, it would be much easier to read

11. Fig 6b to e: Since it is the same scale on the y-axis for all of these 8 plots, I would stack them and make only one panel with two lines: one for the data expressed in Time (h) and one for the data expressed in ZT (h). So again, you do not need to repeat the axis titles. In addition, it is not clear visually if "Col-0" and "prr5 prr7 prr9" only refer to panels b and d or to c and e as well. Having one panel would simplify this part of the figure.

We would like to thank all four excellent reviewers. We are delighted that they view the manuscript positively and we very much appreciate their constructive criticism and attention to detail. We have tried to address all of the comments and give our detailed responses to each of the four reviewers concerns below:

“Reviewer #1 *This manuscript reports on the diurnal oscillation of the coenzyme thiamin diphosphate (TDP) in the nucleus independently of the circadian clock. To prove this, the authors indirectly measured TDP levels in the nucleus of wild-type and circadian clock Arabidopsis mutants by measuring transcript variants of the biosynthesis gene THIC that are under the control of TDP levels via a unique riboswitch TDP sensor. The authors also show the importance of fine regulation of TDP homeostasis on plant’s fitness by controlling the whole plant primary metabolism. They also used Arabidopsis mutants to demonstrate in planta that the mitochondrial TDP transporter TPC is essential for plant’s life, and that the plastidial NSC1 (or PLUTO) transporter is important in the absence of de novo TDP biosynthesis. The manuscript is well written, the methods are scientifically sound, and the conclusions are supported by the results. I only have minor comments:”*

We thank the reviewer for this positive feedback and address the minor comments below.

“Supplementary Fig. 2e: add legends of squares (black = TPC1; white = TPC2)”

Done.

“The authors should explain why they chose the tpc1-1 mutant instead of tpc1-2 mutant in their cross with tpc2-2 mutant. tpc1-2 has an insertion in an exon while tpc1-1 has an insertion in an intron. Insertion in an exon is more likely to produce a complete knock out.”

The *tpc1-1* mutant was deemed suitable for crossing in this experiment due to the severely reduced expression of *TPC1* in this line (see Supplementary Fig. 2c). In principle either of the *tpc1* mutants could have been used. However, we have since noted a mistake in this figure. The cross was done between *tpc1-1* and *tpc2-1* (not *tpc2-2*) conferring phosphinotricin (Basta™) and sulfadiazine resistance, respectively. We apologize for this oversight and have made the appropriate corrections in the text, Supplementary Fig. 2c and Supplementary Table 1. In addition, the use of these lines facilitated segregation analyses due to the different selection resistances (Basta and sulfadiazine).

“Figure 4b: the figure legend mentions that error bars represent SE, but there is no error bar in the graph.”

In fact there was an error bar, it was just not visible because it is so low. We have replotted the data to include a break on the y-axis to make it visible and plotted the standard deviation. We have modified the legend to reflect this and give the p-value as a result of a *t*-test.

“Line 212: “thirteen days after germination” while Fig 4c indicates day 11. Which one is correct?”

We thank the reviewer for pointing this out. 13 days is correct and has been changed in Fig. 4c. Note Figure 4c now includes extra pictures of the seedlings.

“Line 277-279: “and under circadian conditions” – I honestly do not see any rhythm on this graph, except the peak within the first 36 hours.”

Yes, the reviewer is correct--there is no robust rhythm under circadian conditions in this case. We have rephrased this sentence to make it clear.

*“**Reviewer #2:**This is careful work that investigates a set of novel questions relating to vitamin cofactor metabolism in plants. The authors provide important new information demonstrating the importance of B1 vitamers metabolism. They also convincingly demonstrate circadian regulation of the levels of transcripts of the components of this pathway. The study makes technically-demanding measurements of B1 vitamers within timecourse experiments, making this probably one of the first studies to directly investigate the clock control of plant enzyme cofactors. Using an elegant riboswitch reporter, the authors investigate daily changes in some of these molecules at a subcellular scale, which is an interesting finding that also speaks to the broader question of the involvement of circadian regulation in the subcellular compartmentation of metabolism (that broader issue could be emphasized more fully to add impact to the manuscript). There are some places in the article where the level of replication is inadequate to support the conclusions that are reached. The work raises some interesting questions for the future about, for example, the mechanisms of daily regulation of the metabolite rhythms. Could it be driven by photoreceptors or a plastid-based process, for example?”*

We thank the reviewer for the positive comments on the manuscript and are indeed compelled and excited to explore the mechanisms behind the daily regulation of metabolite rhythms. This is an emerging concept in all organisms and provides a fertile future research area for the field. We address the comments dedicated to the manuscript below:

“Major points: 1. At times, the manuscript appears to be two distinct papers that have been combined. Why did authors decide to focus on clock control rather than other responses, such as photosynthetic regulation or stress tolerance? The paper would work better as a single entity if the authors can make provide a more robust justification for why they chose to focus upon daily and circadian regulation within the second part of their article.”

This is a justified comment and we agree that the impetus to study the daily regulation of intracellular TDP supply was not strong enough. So we have now rewritten the first few sentences starting these sections such that it now reads **“Temporal separation of TDP biosynthesis and transport by the circadian clock.** On a daily basis photosynthetic cells need to make dramatic metabolic adjustments during the transition from light to dark to maintain energy supply. As TDP supply needs to be tightly coordinated with the needs of the key enzymes essential for metabolic homeostasis, we were next driven to study the relationship between TDP biosynthesis *de novo* and transport into the organelles over the course of the day. Firstly, we searched the diurnal database....”.

We believe this now provides a more solid validation and link between the first sections on defining the essentiality of TDP for key metabolic processes, the components facilitating its transport and the need to regulate and coordinate biosynthesis and intracellular transport of TDP on a daily basis.

“2. I am confused about the authors’ thinking about the proposed plant NCSI homolog. The authors say that the bacterial ortholog of NCSI might be involved in HMP transport but in the model in Fig. 1, in plants it appears that it is TMP that is transported across the chloroplast envelope instead. Could the

authors add a sentence about the degree of specificity possessed by such a transport protein? Could the transporter have promiscuity for these different related metabolites”

Yes, TMP needs to be exported out of the plastid to be converted into TDP by TPK, which is localized in the cytosol. The transporter for this process is not known currently, as well as the nature of the phosphatase that converts TMP into thiamine. These processes are depicted in gray to illustrate that they are not defined. Nonetheless, the TDP that is made in the cytosol needs to be **imported** into the plastids and mitochondria to facilitate the enzymes dependent on TDP as a coenzyme. The TPC transporters facilitate this process for the mitochondria but the transporter that imports TDP into the plastids is not defined. We have modified Figure 1 to reflect these parameters and the missing/unknown TDP plastid importer is drawn in gray.

NCS1 was described as a nucleobase transporter by Witz et al., and was shown to transport uracil and cytosines. Later, Beaudoin et al noticed that the bacterial homologs of *NCS1* cluster with genes involved in thiamine biosynthesis and salvage therein and provided evidence that it is also, at least, capable of transporting HMP—probably in a salvage pathway. They did not test its involvement in thiamine or TDP transport. Therefore, it is likely to be a promiscuous transporter of several substrates, the specificity and circumstance of which is unknown. In our work, we show that NCS1 is localized to the plastid membrane and is present in shoots but is more highly expressed in roots. To demonstrate the involvement of NCS1 in transport of thiamine, we crossed a *ncs1* mutant line with the TDP biosynthesis *de novo* mutant line *thiC*. The *thiC* mutant alone is fully rescued by thiamine supplementation. However, the *thiC ncs1* double mutant is not fully rescued by thiamine supplementation, demonstrating the involvement of *ncs1* in the transport of thiamine. As NCS1 is localized to the plastid membrane and compromises the rescue of *thiC* mutant plants by supplementation with thiamine, we assume it can either import TDP into the plastid or HMP derived from a salvage pathway that would bypass the need for THIC. We have modified Figure 1 to reflect that NCS1 may import TDP into the plastid. It should be noted that, as TDP import into plastids is essential for plants and as knocking out *NCS1* is not lethal, another (specific) TDP importer must exist but is currently unidentified (also depicted on Figure 1). This latter aspect is already mentioned in the discussion. But, yes, NCS1 is likely a promiscuous transporter that functions in certain tissues under certain conditions, e.g. in roots when the biosynthesis *de novo* pathway is compromised. We have added a sentence on these aspects in the introduction to this transporter in the results section: “Thus, NCS1 may be a promiscuous transporter but the nature of its operation and specificity is currently undefined. Given the implication of *NCS1* in transport of thiamine molecules, we were prompted to look further at this gene in the context of this study.”

“3. How do they know [TDP] oscillation is only nuclear? For a non-specialist reader it would be good to have a simple explanation for why the riboswitch system is detecting only nuclear processes.”

We do not know if the TDP oscillation is *only* nuclear—it may occur also outside the nucleus but we assume that the oscillation that we measure using the riboswitch system is mainly nuclear because the spliceosome is predominantly (if not exclusively) localized to the nucleus. We agree with the proposition of the reviewer and have added a sentence to explain this “As the spliceosome machinery is considered to be predominantly localized to the nucleus, we assume that the riboswitch system reports free TDP levels mainly in this organelle.” We have also modified other sentences as appropriate in the manuscript for consistency.

“4. It might make more sense to combine Supplemental Fig. 3 into Fig. 1. It seems that this supplemental figure presents a very similar data type as Fig. 1a and that the two would fit nicely alongside.”

Agreed. We thank the reviewer for this suggestion, it would indeed fit better in Figure 2a (we assume the reviewer meant Figure 2a rather than Figure 1a—as the latter is the scheme of the pathway). However, Supplemental Fig. 3 is now part of Figure 4c. To address the concern below of this reviewer (and reviewer 4), we have added extra images of *ncs1* and the double mutant *ncs1 thiC* in the presence of various amount of thiamine to support our conclusions about *ncs1* being necessary when biosynthesis *de novo* is compromised. This also includes images of individual seedlings of *thiC*. So they are now all together in one Figure panel (4c) and we have referred to them in the text. Consequently, the old Supplemental Fig. 3 has now been removed.

“5. I do not think the level of replication in Fig. 4c is adequate for any conclusions to be reached from these images. How do we know that the authors are not cherry-picking seedling images that support their hypothesis? It would be much better to quantify some parameters, e.g. cotyledon size or chlorophyll content to provide a proper replicated and quantitative measure. This does not appear to be a well-executed experiment that can be interpreted robustly.”

We now show more images and of three independent crosses of *thiC* with *ncs1* at various thiamine supplementation levels -- these support the conclusion that rescue of *thiC* by thiamine supplementation is compromised in the absence of *ncs1*. We believe that these images of the crosses now show that our data is consistent.

“6a. I am confused about the interpretation of the data in Fig. 6b-e in relation to the data in Fig. 5a. What form of the THIC transcript was quantified in Fig. 5a and how does this compare with the two splice forms that were monitored in Fig. 6? It would be really helpful to have a diagram that illustrates the primer design that was used to detect these 2 or 3 classes of the transcript. Given that the THIC transcript has a rhythm in Fig. 5a, could this alternatively indicate that the rhythms in Fig. 6b are due to a rhythm in the strength of the promoter that is regulating THIC-IR and THIC-IS? Did the authors make or try a regular promoter-luciferase reporter for THIC? If the data in Fig. 6 discount this possibility for THIC-IR and THIC-IS (perhaps the different phases of the two splice forms?), it would be helpful for the authors to provide a robust explanation in the text.”

Fig. 5a is a quantification of the coding region of the gene (CDS), i.e. the beige region in the illustration of Fig. 6a. On the other hand, Fig. 6a quantifies the splice variants of the 3'-UTR region (IS or IR, as depicted). The *THIC IR* version is what is predominantly observed under free-running (circadian) conditions and is therefore the form mostly being detected when monitoring the CDS under circadian conditions in Fig. 5a. A *THIC* promoter driven LUC (without the 3'-UTR of *THIC*) would show the same thing, and has been reported by Bocobza et al 2013 Plant Cell for a *THIC* promoter driven fluorescent protein (RFP). We hope this clarifies the interpretation.

“6b. These data appear to be over-interpreted in places. The authors state, “On the other hand, a robust rhythm was observed for THIC-IS ... under circadian conditions albeit less pronounced”. This is simply not true. THIC-IS does not appear to be rhythmic in the triple mutant under constant conditions. There is no quantitative analysis to support this conclusion and the pattern of peaks in the data do not support this interpretation. The authors appear to use this to reach the conclusion, “it therefore appears that there is oscillation of TDP levels independent of circadian clock function.” This statement is true based on the

data collected in the triple mutant under diel conditions- and well supported by the subsequent luciferase experiments- but cannot be concluded from the data from the triple mutant under constant conditions.”

Yes, the referee’s point is correct, we had written this incorrectly. It has now been corrected in the text as per the comment also from reviewer 1. There is indeed no robust rhythm for THIC IS under constant conditions in the triple mutant. We have rephrased the text accordingly, the rhythm we see for THIC IS is a function of a light-dark cycle.

“Minor points/typos/editorial, General: It would be preferable for the authors to use “diel” rather than “diurnal” when they refer to a process that occurs over a day/night cycle. This is inconsistent within the manuscript. “Diurnal” also means the converse of “nocturnal” so is ambiguous.”

Agreed, we have replaced diel with diurnal throughout the text as appropriate—in addition in some places we now state light-dark cycle to avoid confusion.

“General: For consistency, it is recommended to be careful about interchanging “gene expression” and “transcript levels”. “Transcript levels” is probably more accurate when the data show transcript levels.”

Agreed, we have replaced gene expression with transcript levels when the data show transcript levels.

“General: The authors say “data not shown” at several points in the manuscript. Why have they excluded these data? If they are negative results (e.g. with NCSI in yeast), why not include them because they are informative?”

We have now included the yeast data. In particular, attempts to complement the yeast strain deficient in transport and biosynthesis of thiamine with *NCSI* are included as Supplementary Figure 3. We have also included a section in the methods on these experiments.

“General: A number of the figures lack scale bars and, depending on journal policies, these might be needed. For example, Supplemental Fig. 3 might need scale bars for a valid comparison of seedling size.”

Scale bars are now added where appropriate including Fig. 4c, which now incorporates what was Supplemental Fig. 3 before.

“General: It would help readers to have referencing of the figures within the Discussion text.”

Agreed, we now refer to the figures in the Discussion.

“General: I find the use of the phrase “clock autonomous metabolic oscillations” to be obscure. Do the authors mean oscillations of metabolic processes that occur independently from the known circadian oscillator, and/or remain rhythmic in the absence of a functional transcription-translation circadian oscillator? If so, I do not think the authors have identified “clock autonomous metabolic oscillations” because the oscillations they have measured appear to be driven by the light/dark cycle.”

Yes, we agree this is obscure. We mean oscillations of metabolic processes that are independent of the known circadian oscillator and have clarified this where mentioned in the text (i.e. the abstract).

“Fig. 2a; it would be helpful if the authors could provide some quantification of a factor within their images to provide readers with a sense of the quantitative differences and variation in individual seedlings of this response. Could they extract a parameter (such as mean visible plant area per seedling?) for each thiamine concentration? Perhaps the wild type is a bit bigger at 1.5uM thiamine, but it’s hard to tell without some quantification.”

In line with the earlier suggestion to integrate Supplementary Figure 3 into Figure 2, we now show representative images of individual seedlings grown under the different thiamine concentrations in Fig. 4c. The differences with respect to thiamine supplementation are much clearer with the individual seedling images.

*“Fig. S2d; given that metabolite concentrations can be variable, it would be sensible for the authors do perform some statistical analysis on these data. For example, is [TDP] lower in *tpc1-1* than the wild type? Alternatively, it could be informative to overlay a dotplot or swarm plot.”*

There was no statistical difference for the individual vitamins across the lines. We have now added this information in the legend.

“Fig. S2e; this seems to be a confusing presentation of the data. Would it not be clearer to divide the graph into two parts, one comparing TPC1 across the developmental stages and anatomy, and one beside this for TPC2?”

This may have been confusing because we omitted to add the legend of the squares (black = TPC1; white = TPC2)—it is now included. We prefer not to separate the graph, as we believe that it is easier to compare the expression of both genes across the parameters measured when both are on the same graph.

“Fig. 4b; the authors state in the figure legend, “The data are the average of three independent biological replicates with error bars representing SE” yet no such error bars are present on the graph and there is no explanation of any statistical analysis that led to the starring on the plot.”

Already dealt with in line with a similar comment from reviewer 1. In fact there is an error bar, it is just not visible because it is so low. We have replotted the data to include a break on the y-axis to make it visible and plotted the standard deviation. We have modified the legend to reflect this and give the p-value as a result of a *t*-test.

“Fig. 5a; why is the transcript level of CCA1 so variable in the triple mutant? Is this expected? This should at least be commented upon briefly within the Results section.”

The level is not highly variable, there are two time-points (ZT 56h and 84h) that do not adhere to the general trend of increasing CCA1 levels. This trend is expected because the repressors of CCA1 expression (i.e. PRR5, 7 and 9) are absent in the *prr975* mutant. A log plot hides these discrepancies but we have specifically chosen not to show log plots for this reason. In any case, we have added a sentence on this in the results section “Notably, the tendency for an increase in CCA1 transcript levels in *prr5 prr7 prr9* is consistent with the known repressor function of the PRR5, 7 and 9 proteins on CCA1 expression (Fig. 5a).”

“Line 97-98; “in order to demonstrate that” -> “in order to test whether” [the former suggests a lack of scientific equipoise on behalf of the authors].”

Agreed and done.

“Line 110-111; does this statement need a caveat that the concentration in certain subcellular compartments might be greater or lower than this approximate threshold of 1500 pmol g-1 FW?”

We have added “at the tissue level” to this statement in the text

“Line 122; when discussing these transport proteins, it is important to explain in the text that At3g21390 is distinct from the two-pore channel TPC1 transport protein (At4g03560). I make this comment particularly because At3g21390 is not yet annotated as TPC1 within the TAIR/Araport databases, whereas the TPC1 Ca²⁺ channel is already annotated. At least with Arabidopsis, At3g21390 should possibly be annotated with a distinct name from TPC1. A sentence of clarification would rule out any potential ambiguity by the readership.”

Agreed, we have now included a sentence of clarification in this section of the text. “Note, Arabidopsis TPC1 and TPC2, should not be confused with the Arabidopsis two pore channel (TPC) protein at locus At4g03560.”

“Line 142; the authors should provide very brief justification of why they chose specific alleles of each gene for their crosses”

Already dealt with in line with a similar comment from reviewer 1 (see above).

“Line 147; please can the authors expand upon, “a sole embryo based problem”? What does this mean? Could the result suggest a male- or female-specific germline effect?”

Yes, it could also be a germ-line effect; we have added this information to the text.

“Line 171; what is revealed by informatic tools about the putative transit peptide? Such an analysis might add confidence to this interpretation. If it appears that the transit peptide is, on probability, a chloroplast transit peptide then could the results with the yeast complementation experiment be simply interpreted in terms of the fact that yeast doesn’t have chloroplasts and therefore the metabolic compartmentation is different in yeast?”

We have now added the information that the informatics tools (iPSORT, ChloroP, TargetP) predict a chloroplast transit peptide.

“Line 179 onwards; no information is provided here or in the methods about the size of the promoter fragment used for the GUS assay. This is essential for reproducibility of the work.”

A fragment of 519 bp upstream of the start codon of NCS1 was used. This information is now included in the methods section.

“Line 179; the claim about dotted GFP signal being consistent with plastid localization is over-interpretation in the absence of a known plastid marker. It would be better to say that based on the data in Fig. 3a, the most reasonable interpretation is that this is indicative of plastid localization.”

Agreed, we have now rephrased the sentence as suggested.

“Line 263; there is no reason “free” needs to be in italic.”

Agreed, it is now in plain font.

“Line 257; “changes in the overall contents...” of exactly what?”

B₁ vitamers. We have now added this to the text.

“Line 302; “While regulation of...” this sentence seems garbled or over-edited. Perhaps divide it into two sentences?”

Agreed, we have divided it into two sentences.

“Line 331; “Here we also show that both biosynthesis and transport of TDP are regulated at transcriptional level by the circadian clock.” – this might be over-interpretation because the authors do not have information about the activity of the enzymes encoded by these genes. A slightly different form of words would be more parsimonious.”

We agree, which is why we have said “at the transcriptional level”.

*“**Reviewer #3:**In this paper by Noordally et al, “The coenzyme thiamine diphosphate displays a nuclear rhythm independent of the circadian clock,” the authors explore the role of transporters in thiamine mobility in planta and the effect of daily light cycles on the post-translational regulation. The authors report on the changes in metabolite levels in wild type and thiC mutants in response to supplemental thiamine. The authors report on the lethality of tpc1/tpc2 double mutants, and an interaction of ncs1 with thiC during recovery with supplemental thiamine. The authors also report on the subcellular and tissue specific expression of NCSI. The authors also report on the expression of multiple thiamine metabolism genes and the TPCs where they extrapolate that transport maybe temporally separated from circadian peak, however, changes in TPC1 protein levels are not shown. Finally, the authors explore diurnal vs. circadian peaks for the ribo-switch mediated splicing and observe a diurnal peak of splicing as opposed to the nocturnal peak of expression. This paper is a good start, but incomplete story. Many processes can be rhythmic without clock control, and metabolism is no exception. The major finding is a single panel that splicing has a daily rhythm that is presumably light responsive. However, the authors do not explore this result further. The authors have to show much more for this to be of interest to a diverse audience. I.A main thrust of this paper is the report that proper post-transcriptional control of a single transcript in the thiamine is driven by the environment, not the clock. Although many forward the idea the metabolic rhythms are influenced by the clock, this idea is far from “the central dogma is that metabolic rhythmicity is a clock driven process” as described in the abstract. This is an overstatement and should be removed from the abstract. Most processes in plants are influenced by the environment (light/dark cycles, temp cycles, rainfall cycles, humidity cycles) and are rhythmic, but independent of the clock, including metabolism. The observation of post-translational regulation with a daily, non-circadian clock driven oscillation is not striking. The authors should reconsider the abstract.”*

We have removed the statement that “the central dogma is that metabolic rhythmicity is a clock driven process” from the abstract. However, we would like to emphasize that the main thrust of this paper concerns the fact that no rhythm of the coenzyme thiamine diphosphate (TDP) can be observed at the tissue level (unlike many metabolites). One has to look at the subcellular level to see a rhythm for this metabolite/coenzyme. In this particular case, we could make use of the TDP riboswitch to demonstrate this intracellular metabolite rhythm. To our knowledge this has

not been shown before for any coenzyme and is appreciated by the other 3 reviewers. This TDP rhythm at the subcellular scale is likely a result of the supply and demand of TDP at this scale, i.e. TDP, which is made in the cytosol, must be supplied to the enzymes that depend on it within organelles (plastids and mitochondria), and importantly when they need it to achieve metabolic homeostasis. This is mediated by biosynthesis and organellar transport of TDP and is further managed by the riboswitch to ensure metabolic homeostasis. We are not simply describing a metabolite rhythm, nor invoking any idea that it is regulated at the post-translational level. This is a post-transcriptional process coordinated in the nucleus, mediated by the only riboswitch known in plants.

“2.The results on NCS1 are very preliminary and not quantified. The authors should quantify the thiamine rescue of the ncs1 thiC double mutant (FW biomass, or other measurement) compared to thiC alone. Qualitative reports based on single picture is insufficient. Also, the authors should complement mutant line to show that the deficiency in growth (if significant) is due to loss of NCS1.”

We now show more images and of three independent crosses of *thiC* with *ncs1* at various thiamine supplementation levels -- these support the conclusion that rescue of *thiC* by thiamine supplementation is compromised in the absence of *ncs1*. We believe that the images of the crosses show that our data is consistent. While complementation could have been done, we have not done it here because we believe our point was made by showing that a phenotype is observed with the double mutant *thiC ncs1*, and because *ncs1* alone has no distinguishable morphological phenotype, i.e. *ncs1* is only needed when thiamine biosynthesis *de novo* is compromised.

“3. The authors may want to consider testing if NCS1 is a TPP transporter by targeting it to the mitochondria to see if it can complement a tpc1/tcp2 double mutant. Similarly, the authors could also show that TPP transport is the defect in the ncs1 mutant by targeting a tpc to the chloroplast membrane. These experiments could illustrate that the transporters are indeed moving similar metabolites across membranes.”

As described in the manuscript the *tpc1 tpc2* mutant is not viable. We have commented on the fact that NCS1 is a promiscuous transporter and appears to function in thiamine transport only when biosynthesis *de novo* is compromised, i.e. the bona fide plastid transporter (importer) remains unknown (see above comments to reviewer 1 and 2). Therefore, while the suggestions of the reviewer have merit, the approach would be better suited to prove the functionality of the bona fide plastid TDP importer once identified.

“4.The authors forward the idea biosynthesis and transport are temporally segregated processes, however, the authors old describe the transcription of the genes, and do not show that either biosynthesis or transport are changing over the day. In fact, Figure 5B shows that major products of the thiamine pathway are constant over a 24 hour day.”

See reply to point 1, the reviewer may have missed the main thrust of this paper and that the TDP rhythm pertains to the subcellular rather than tissue scale.

“5.Although the authors forward the idea that there is a daily rhythm of TPP levels as seen from the splicing of the ribo-switch as part of a luciferase reporter construct (Fig 6B), the daily levels of TPP do not change (Figure 5B). How are TPP levels in the nucleus connected to the lack of daily rhythms in the metabolite itself. This is a big disconnect. One possibility is that alternative splicing factors are regulated

by light, as is being reported (<https://doi.org/10.1073/pnas.1407147112>, <https://doi.org/10.1093/pcp/pcy089>). It could be that TPP levels are constant (as seen in figure 5B) but the coincidence with light triggers intron retention vs. splicing.”

See reply to points 1 and 4. Moreover, we have recently shown that sensing of TDP through the riboswitch is essential for metabolic homeostasis in Rosado-Souza, L et al 2019 Plant Physiology, and supports our postulations here as indicated in the discussion section.

“6.The authors show primers and experimental design relative LUC-IR transcripts, but presumably no products are observed since they would require splicing to bind. The authors should design primers similar to those for testing THIC-IR to determine how does the expression of LUC-IR to LUC-IS compare. The authors should quantify the abundance of the spliced vs. un-spliced versions to each other. Also, does the LUCIFERASE reporter (by bioluminescence, as the authors report using to identify the transformed lines in the materials and methods) reflect the IR or IS confirmation in diurnal conditions?”

We thank the reviewer for pointing this out. We should not have illustrated binding sites for LUC-IR because we only measure LUC-IS in this experiment. The reason for this is that we showed in parts a-e of this figure that THIC-IR responds robustly and probably exclusively to the circadian clock, whereas THIC-IS directly responds to the levels of TDP. THIC-IS oscillates in light-dark cycles, even in an arrhythmic mutant but not in free-running (circadian) conditions. Therefore, THIC-IS oscillation is a function of light-dark cycles and not the circadian clock. To prove this further, we used the LUC construct that is under the control of a constitutive promoter, thereby removing any control by the circadian clock. We only need to monitor LUC-IS because this spliced version is directly and only responding to TDP levels and will tell us if there is an oscillation—which was the case in light-dark cycles but not under free-running conditions, i.e. not circadian driven.

Bioluminescence was not sensitive enough to see the changes at this subcellular level, and is why we used qPCR to monitor the spliced version.

“7.Does THIC-IS show an ultradian rhythm in LL (Figure 6c)? It seems that there is a double peak, one during the day following the diurnal peak and another at night when the unspliced transcript peaks. One might be observing the intersection between a circadian-regulated daily peak of transcript expression and post-translational splicing. Higher time resolution might resolve this (2 hr resolution).”

We do not think this is an ultradian rhythm *per se*, but is simply “noisy” due to the overall *THIC* transcript abundance being under the influence of both the circadian clock, the alternative splicing due to TDP levels i.e. the two events are coupled and all in combination with the stability of the *THIC* transcripts themselves. These events start to become uncoupled in the arrhythmic clock mutant, where we robustly see the rhythm of TDP itself under light-dark conditions.

“Minor criticism: Line 24: “we then took advantage of the sole riboswitch metabolite sensor in plants...” Please include “identified so far” or “sole reported” as the search for riboswitches has not been exhaustively performed.”

Agreed. We have added “sole reported”.

“Fig 2c: bottom legend for color chart should read ≤ -2 ”

We thank the reviewer for pointing this out. It is now corrected.

“Reviewer #4: Review on “The coenzyme thiamine diphosphate displays a nuclear rhythm independent of the circadian clock” by Noordally et al. Noordally et al. explored in Arabidopsis metabolic oscillations of the thiamine diphosphate (TDP), the form of vitamine B1 acting as a coenzyme and that plays important role in carbon metabolism. The authors showed that genes involved in TDP biosynthesis and transport are transcriptionally regulated by the circadian clock and that these two mechanisms are temporally separated at transcript level. Finally, they suggested that oscillations of TDP detected in the nucleus are independent of the clock function. Noordally et al. performed an interesting work on clock autonomous metabolic oscillations, an original field that is still poorly understood in plants. I appreciated the originality of this work, the clarity of the figures and the interesting discussion. However, some conclusions in the result section are too direct and should be tone down, or additional data should be provided to support these conclusions. I also have a few comments/suggestions to improve the figures. Particularly, scale bars are missing on all pictures and should be added.”

We thank the reviewer for the very positive comments and are delighted that the originality of the work is appreciated. Specific concerns are addressed below.

“Major points:Introduction:1. I am not sure it is necessary to describe the mechanisms in the figure legend (Figure 1), but I am ok with that. However, the role of CCA1 in the regulatory processes should be explained in the introduction, not only in the legend.”

Agreed, we have now included it also in the introduction.

“Results:2. l.100: “Supplementation with thiamine had little effect on wild type growth (Fig. 2a). However, the growth of thiC beyond the cotyledon stage was dependent on the level of thiamine supplementation (Fig. 2a).” and l.110: “it appears that ca. 1500 pmol g⁻¹ FW is sufficient to sustain growth at wild type levels (Fig. 2b)”I agree with the authors that thiC growth depends on the level of thiamine. However, based on the plant pictures, it seems like the 1.5 concentration do not completely rescue the phenotype observed for Col-0, while levels of TDP seem to be similar in the two genotypes under this thiamine concentration. This result needs phenotypic data to confirm the conclusion, such as number of leaves per plant, leaf surface area, fresh mass, dry mass, etc., for each thiamine concentration. Otherwise, the conclusion should be significantly tone down and the authors should discuss the fact that high TDP levels in plants may not be sufficient to completely rescue the thiC phenotype. In addition, statistical results should be added to confirm or not that TDP levels and plant phenotypic variables are similar between the two genotypes under 1.5 μM of thiamine.”

In line with a similar comment of reviewer 2 and another relating to additional images of *ncs1* and *ncs1 thiC*, we have now integrated Supplementary Figure 3 into Figure 4 (part c), such that it now shows representative photos of individual seedlings under various thiamine supplementation levels. It can be seen that at 1.5 μM thiamine, *thiC* is very similar to wild type. Nonetheless, we agree with the reviewer and have significantly toned down the conclusions, as we do not have quantitative data of the suggested parameters for each thiamine supplementation level. It now simply reads “From the measurements of TDP levels in these seedlings, it appears that ca. 3800 pmol g⁻¹ FW (at the tissue level) is required for growth to approach wild type levels (Fig. 2a and b), because stunting and chlorosis can still be observed with lower supplementation levels (Fig. 2a; see also Fig. 4c).”

“3. l.132: “However, no morphological defects could be deciphered compared to the wild type (Col-0)

grown under the same conditions, suggesting functional redundancy of TPC1 and TPC2 (Supplementary Fig. 2b). "Unless it is known that tpc1 and tpc2 mutants alter plant phenotype, I would remove the second part of the sentence in the absence of a double mutant phenotype. To me, this result does not necessarily suggest a functional redundancy between the two genes. Moreover, the authors demonstrated that TCP1 is cycling under constant light, while TCP2 does not exhibit circadian oscillations (Fig 5a), suggesting that these two genes are not similarly regulated."

Agreed, we have removed the second part of this sentence.

"4. Scale bars should be added in: - Fig 2a: - Fig 3a, c and d- Fig 4c"

We have now added scale bars for Fig. 2a, Fig. 3a and d (we do not think it is necessary for part c). Fig 4c (see above) has changed but we have added a scale bar nonetheless.

"5. Based on published Chip-Seq data, THIC and TH11 are both targeted by CCA1 and LHY, and NCS1 is targeted by PRR9 (see Nagel et al, 2015; Kamioka et al, 2016; Adams et al, 2018; Liu et al, 2016). This information should be added in the text and would add strength to the authors' conclusion on l. 244."

Agreed and added.

"6. l. 294-295. "...we therefore conclude that there is an oscillation of the TDP metabolite in the nucleus that is independent of circadian clock function" I understand and I agree with this interpretation. However, I am surprised that the authors did not discuss the role of CCA1 in this paragraph, while this gene is part of the core clock machinery and is the main regulator of THIC expression. Also, I think that expression profiles of CCA1 in both LD and LL conditions and in both Col-0 and prr5-prr7-prr9 triple mutant should be added to this figure 6. In prr mutants, oscillations of CCA1 are disturbed, but the gene is still expressed (for example, Salome et al, 2010, The Plant Cell, Fig 2A). This probably explains why THIC-IS transcript is still produced in the triple mutant under constant light (Fig 6e), but this transcript (THIC-IS) is not cycling probably because TDP is not cycling under constant light, as described by the authors. So, I think adding CCA1 expression profiles would add strength to the conclusion and would help the reader to understand this interpretation. Again, the role of CCA1 should be discussed more in the manuscript."

The profile for CCA1 in LL conditions in both Col-0 and *prr5 prr7 prr9* is in Fig. 5a. We have already commented on the behavior of CCA1 in the triple mutant in the context of a comment from reviewer 2: "Notably, the tendency for an increase in CCA1 transcript levels in *prr5 prr7 prr9* is consistent with the known repressor function of the PRR5, 7 and 9 proteins on CCA1 expression (Fig. 5a)." Yes, the reviewer is correct, TDP does not cycle under constant light—i.e. light/dark transitions are required for the cycling to be seen (which is what we show).

"Minor points:1. Arrow color code should be explained in the legend of Figure 1. I guess that the blue arrow represents the "negative feedback regulation on TDP biosynthesis de novo" caused by the "binding of TDP to the THIC riboswitch", as mentioned in the introduction (l. 74-76). This should be clarified in the legend."

Agreed and done.

"2. Fig 2b: More numbers on the x-axis should be added."

Agreed and now added.

“3. Fig 2c: Please indicate "thiamine supplementation" as a y-axis title and add numbers on the y-axis, remove text at the top of the plot and add a legend for the dot size.”

Agreed and now added.

“4. Supplementary Fig 1: This figure is highly informative. Maybe I missed it, but I think this data should also be provided as a Supplementary Table, with Pvalues and log2FC. Also, a heatmap of the log2FC values for all significant metabolites (one column for a and one for b) would allow us to easily compare metabolite responses in the two genotypes. This could be Supp Fig 1c.”

As our aim here is to illustrate the overall impact of thiamine deficiency on general plant metabolism, we feel it is unnecessary to supply all of the data in these different formats. The metabolites that have significant changes are illustrated in Fig. 2c.

*“5. Supplementary Table 1: Please check the table legend "($P > 0.05$; * $P > 0.05$)”.*

Yes, thanks to the reviewer for this. It is now written correctly.

“6. l.212: "...more apparent thirteen days after germination": 13 days or 11 as indicated on Fig 4c? Please clarify.”

We thank the reviewer for pointing this out (also noted by reviewer 1). 13 days is correct and has been changed in Fig. 4c. Note Figure 4c now includes extra pictures of the seedlings.

“7. From DIURNAL, data obtained in LL condition are available and could be plotted in Supplementary Fig 4 as well.”

Agreed and added.

“8. Fig 5a: A legend for the two genotypes would facilitate the understanding without the need to read the figure legend.”

Agreed and included.

“9. Fig 5a: This panel could be expanded a little bit if you do not repeat the axis titles for each individual plot”

Agreed and done.

“10. Fig 5b: If you do not repeat the axis titles and if you indicate the measured compounds at the top of each individual plot, it would be much easier to read”

Agreed and done.

“11. Fig 6b to e: Since it is the same scale on the y-axis for all of these 8 plots, I would stack them and make only one panel with two lines: one for the data expressed in Time (h) and one for the data expressed in ZT (h). So again, you do not need to repeat the axis titles. In addition, its is not clear visually if "Col-0" and "prr5 prr7 prr9" only refer to panels b and d or to c and e as well. Having one panel would simplify this part of the figure.”

We now indicate “relative transcript abundance” just once at the side, instead of repeating it. We would prefer not to stack the plots, as we think this figure is already busy and would make it difficult to read. We prefer to have Col-0 and *prp5 prp7 prp9* in separate panels but we have indicated the appropriate titles on all 4 panels (as this was omitted before) to clarify which line was being analyzed.

REVIEWERS' COMMENTS:

Reviewer #1 (Remarks to the Author):

The authors addressed all the points I raised satisfactorily.

Reviewer #2 (Remarks to the Author):

I enjoyed reading this revised manuscript, where the authors have made considerable effort to address the concerns I raised regarding the original submission. The manuscript is now much improved. I have read the changes carefully, and re-read the manuscript overall. I am very positive about this study, and whilst reading the manuscript noticed some issues that the authors might want to consider further. I do not think these points preclude publication of the work, but if they can be addressed it will make the paper even better:

1. I still do not think the authors have made clear enough within the Introduction why they chose to investigate circadian regulation in the context of TDP metabolism. The context for why circadian rhythms were investigated is a little disappointing. It is very clear from the abstract, whereas the Introduction doesn't justify this well. The Introduction presents nice background to TDP metabolism, and ends with a paragraph to the effect of "by the way, we looked at clock control also." This needs to be better justified. It would not be unreasonable to mention circadian regulation earlier in the introduction also, to set up those experiments in the minds of the readers.

2. (i) The description of how the quantitative analysis of the timecourse data was performed is confusing. The Methods mentions several algorithms that were used (line 539), but the plots in Fig. 5 show a single measure of period and phase. It's not explained in the legend either. In the interests of reproducibility, exactly what algorithms was used to obtain these values? (ii) If the authors have the data, I would encourage them to include a measure of the quality of fit of the analysis to the data. This can help readers interpret the reliability of the period/phase estimates.

3. In Fig. 4C, it seems that the images of the plants are spliced together. When examining the figure, the gaps between the images didn't display or print well (see attached example). I would recommend that the authors increase the spacing between each of the images, first to make clear that it is an assembled image rather than a single photograph with multiple plants, and second so that it displays and prints properly.

5. There is an interesting feature of Fig. 6g that the authors might want to think about/ expand on, or consider for the future. Under light/dark conditions, promoter activity increases during the light period. Under constant light, promoter activity becomes low, when one might expect promoter activity to become continuously high (equivalency to the light of light/dark conditions). Therefore, there is a requirement for a dark period for normal daily cycles of TDP metabolism. Perhaps this seems normal to the authors, but it does seem to be an interesting observation because it might suggest there could be photoperiodic effects, etc. In future work, if the authors wanted to see if the clock has a role in the oscillation in Fig. 6g that occurs under light/dark cycles, the authors could vary the T cycle and track the behavior of the circadian phase (see Fig. 1C in Merrow & Roenneberg Cold Spring Harb Symp Quant Biol 2007. 72: 279-285). It could be worthwhile because of the way that the prr579 mutant seems to have an effect under light/dark cycles (Fig. 6b/d).

Minor

Lines 28/29; the sentence about fitness and adverse settings is peculiar. It suggests the authors think that circadian rhythms are only important under conditions of stress. Results such as those in Graf et al. PNAS 2010 seem to suggest otherwise. I suggest rewording.

Line 40; English is a bit clumsy.

Line 100; figures are not called out in correct order.

Line 122; Refers to proteins, should not be italic.

Reviewer #4 (Remarks to the Author):

Changes made by the authors have significantly improved the manuscript.

The authors responded clearly and convincingly to the comments. From my point of view, I don't see any reason to reject the manuscript.